# Ratiometric fluorescent sensing of pyrophosphate with *sp³*-functionalized single-walled carbon nanotubes

Simon Settele [1], C. Alexander Schrage [2], Sebastian Jung [2], Elena Michel[3], Han Li [4,5], Benjamin S. Flavel [4], A. Stephen K. Hashmi[3,6], Sebastian Kruss[2,7] ✉ & Jana Zaumseil [1] ✉

Inorganic pyrophosphate is a key molecule in many biological processes from DNA synthesis to cell metabolism. Here we introduce *sp³*-functionalized (6,5) single-walled carbon nanotubes (SWNTs) with red-shifted defect emission as near-infrared luminescent probes for the optical detection and quantification of inorganic pyrophosphate. The sensing scheme is based on the immobilization of $Cu^{2+}$ ions on the SWNT surface promoted by coordination to covalently attached aryl alkyne groups and a triazole complex. The presence of $Cu^{2+}$ ions on the SWNT surface causes fluorescence quenching via photoinduced electron transfer, which is reversed by copper-complexing analytes such as pyrophosphate. The differences in the fluorescence response of *sp³*-defect to pristine nanotube emission enables reproducible ratiometric measurements in a wide concentration window. Biocompatible, phospholipid-polyethylene glycol-coated SWNTs with such *sp³* defects are employed for the detection of pyrophosphate in cell lysate and for monitoring the progress of DNA synthesis in a polymerase chain reaction. This robust ratiometric and near-infrared luminescent probe for pyrophosphate may serve as a starting point for the rational design of nanotube-based biosensors.

SWNTs are a promising platform for spectroscopic sensing in the second biological window (NIR-II, 1000–1700 nm) as their optical properties are very sensitive to their environment[1-4]. SWNTs can be seen as rolled-up sheets of graphene with different roll-up angles and diameters that lead to different nanotube species, i.e., chiralities. Each species of SWNTs has characteristic optical properties and exhibits narrow photoluminescence (PL) peaks within the NIR-II window[5]. In combination with their high photostability and biocompatibility, SWNTs are an excellent material for the development of biosensors[6-8]. Over the last decade various SWNT-based biosensors have been developed that are sensitive to, for example, bacterial motifs[9], reactive oxygen species[10], metal ions[2,11], proteins[12], and neurotransmitters such as dopamine[13]. One drawback of SWNTs is their typically low PL quantum yield (PLQY) (<1%) in aqueous dispersion and the low purity of the raw material. To tackle these issues, the field has recently expanded toward the use of sorted monochiral (i.e., single species) SWNTs and the integration of luminescent *sp³* defects, also named quantum defects[14-17].

The intentional introduction of *sp³* defects by covalent functionalization has been shown to enhance the fluorescence properties of SWNTs and increase their PLQY[18-20]. At low densities, *sp³* defects lead to new and bright emission bands (typically labeled as $E_{11}^*$) that are red-

[1]Institute for Physical Chemistry, Universität Heidelberg, Heidelberg D-69120, Germany. [2]Department of Chemistry and Biochemistry, Ruhr-Universität Bochum, Bochum D-44801, Germany. [3]Institute for Organic Chemistry, Universität Heidelberg, Heidelberg D-69120, Germany. [4]Institute of Nanotechnology, Karlsruhe Institute of Technology, Kaiserstrasse 12, Karlsruhe D-76131, Germany. [5]Department of Mechanical and Materials Engineering, University of Turku, Turku FI-20014, Finland. [6]Chemistry Department, Faculty of Science, King Abdulaziz University, Jeddah 21589, Saudi Arabia. [7]Biomedical Nanosensors, Fraunhofer Institute for Microelectronic Circuits and Systems, Duisburg D-47057, Germany. ✉e-mail: sebastian.kruss@rub.de; zaumseil@uni-heidelberg.de

shifted from the native excitonic $E_{11}$ emission. They provide additional fluorescence signals at different wavelengths and with different responses to changes in the nanotube environment and to analytes. Thus, they enable multimodal and ratiometric detection schemes. For the easily sorted and purified species of (6,5) SWNTs (diameter 0.76 nm), the $E_{11}$ emission occurs at $\approx 990$ nm and the $E_{11}^*$ emission at $\approx 1140$ nm in aqueous dispersion[15]. Very recently such $sp^3$ defects were successfully used by Kim et al. as fluorescent probes to detect ovarian cancer and employed by Spreinat et al. to sense dopamine[21,22]. Currently, most detection strategies depend on a complex interplay of the target analyte with non-covalent biopolymer-SWNT hybrids (e.g., ssDNA-wrapped SWNTs) that induce changes in the chemical or dielectric environment of the nanotube. However, the sensing mechanism often relies on very weak interactions and frequently requires complex analysis based on a large number of data sets. More direct detection schemes typically require a more specific interaction of the analyte with the SWNT. As the chemical moieties and functional groups attached to $sp^3$ defects can be tailored, they enable the rational design of targeted analyte bindings and signal transduction schemes. One recent example is the application of SWNTs functionalized with pH-responsive $N,N$-diethylamino moieties. They enhance the optical response to small changes in lysosomal pH and indicate autophagy-mediated endolysosomal hyperacidification in live cells through a shift of the defect emission wavelength[23]. Thus, careful design of sensing schemes using controlled interactions of analytes with the SWNT surface and covalently attached functional groups could facilitate the optical detection of important biomarkers that were previously out of reach. One of these biomarkers is inorganic pyrophosphate.

Inorganic pyrophosphate ($PP_i$, $P_2O_7^{4-}$) plays a critical role in biological systems[24,25]. It is one of the main byproducts of biochemical reactions such as DNA and RNA synthesis and hydrolysis of adenosine triphosphate (ATP) within cells[25,26]. Hence, it is closely related to biological energy storage processes and has become an important biomarker for measuring telomerase activity for cancer diagnosis[27]. Additionally, excess $PP_i$ may promote diseases related to bones and joints. High levels of $PP_i$ are observed in the synovial fluid of patients with calcium pyrophosphate dihydrate (CPPD) crystals, bone attrition and chondrocalcinosis[28,29]. Thus, the detection and quantification of $PP_i$ is highly desirable and the development of corresponding probes has been the subject of extensive research in recent years[30–32].

Fluorescent probes are often used for biomarkers due to their fast response and quantitative real-time readout as well as the potential to use them for in vivo imaging. Water-soluble fluorescent probes for $PP_i$ typically rely on metal displacement assays, in which an acceptor molecule is attached to a fluorophore that switches between an emissive on and dark off state depending on the reversible metal ion chelation (e.g., $Fe^{3+}$, $Zn^{2+}$, $Cu^{2+}$) of the acceptor[31,32]. While this approach can reach sensitivities down to the nanomolar level as well as high selectivity in the presence of other phosphates, the emitted light is typically restricted to the visible (400–700 nm) or at best NIR-I window (700–1000 nm)[31,33]. In recent years, in vivo imaging within the NIR-II has emerged as a method that benefits from ultralow light scattering and deeper penetration through biological tissues[34,35]. Various NIR-II fluorescent probes have been developed, including organic dyes[36,37], gold nanoclusters[38], quantum dots and lanthanide nanocrystals[39–41]. However, they often suffer from limited biocompatibility due to the presence of toxic transition metals (e.g., Pb, Cd) or limited stability due to photobleaching[42,43].

Here, we introduce the direct and quantitative detection of $PP_i$ with tailored $sp^3$-functionalized (6,5) SWNTs as fluorescent probes in the NIR-II window. Sorted (6,5) SWNTs are functionalized with luminescent defects bearing an alkyne moiety and exhibit a high sensitivity towards the presence of $Cu^{2+}$ ions, resulting in strong quenching of the $E_{11}$ and $E_{11}^*$ emission. The quenching effect is reversed by the addition of copper-complexing analytes such as $PP_i$, which can be monitored quantitatively by several different spectroscopic metrics. The intensity ratio of the defect-induced $E_{11}^*$ emission to the $E_{11}$ emission enables ratiometric and thus the most robust detection of $PP_i$. After exploring the PL quenching and thus $PP_i$ detection mechanism, we show that biocompatible, phospholipid-polyethylene glycol-stabilized SWNTs with $sp^3$ defects can be used for reliable $PP_i$ quantification even in complex biological media (e.g., cell lysate) and for fast optical detection of $PP_i$ released during DNA synthesis in a polymerase chain reaction (PCR) as a potential application.

## Results

### $sp^3$-functionalization of SWNTs and fluorescent probe design

To create $sp^3$-functionalized SWNTs capable of detecting $PP_i$, (6,5) SWNTs were sorted via aqueous two-phase extraction (ATPE) and transferred into an aqueous dispersion with the surfactant sodium dodecyl sulfate (SDS) as reported previously[44]. Subsequently, the nanotubes were $sp^3$-functionalized by the addition of appropriate aliquots of 4-ethynylbenzene diazonium tetrafluoroborate and stored for 7 days under the exclusion of light to ensure full decomposition of the diazonium salt (see Methods for details). PL spectra of the obtained covalently functionalized (6,5) SWNTs with 4-ethynylbenzene moieties (referred to as Dz-alkyne) displayed a red-shifted $sp^3$ defect-induced emission feature ($E_{11}^*$) around 1135 nm in addition to the original $E_{11}$ emission at 988 nm (see Fig. 1). A 1:1 mixture of copper (II) sulfate pentahydrate ($CuSO_4(H_2O)_5$) and tris(3-hydroxypropyl-triazolyl-methyl) amine (THPTA) was added to the dispersion of functionalized (6,5) SWNTs. $CuSO_4(H_2O)_5$ and THPTA form a well-defined $Cu^{II}$-complex (from here on labeled as {Cu}) that is frequently used in organic chemistry[45]. Upon addition of {Cu}, both the $E_{11}$ and $E_{11}^*$ emission greatly decreased and the emission peaks red-shifted by 7 and 5 nm, respectively (Fig. 1c). Importantly, the $E_{11}^*$ emission was reduced more strongly, which led to an overall decrease of the $E_{11}^*/E_{11}$ PL intensity ratio (see Supplementary Fig. 1). PL quenching was observed at concentrations of {Cu} above 0.34 μM and became stronger with increasing concentrations of {Cu} until $E_{11}$ and $E_{11}^*$ PL reached a stable level for {Cu} concentrations of >23 μM. Similar behavior for the $Cu^{2+}$ induced quenching of nanotube fluorescence was previously observed by Wulf et al., yet no red-shift in peak position was reported in that study[46]. For high {Cu} concentrations (e.g., 30.4 μM) the $E_{11}$ peak intensity decreased by a factor of $\approx 2$ while the $E_{11}^*$ peak intensity decreased by a factor of $\approx 5$. In subsequent experiments, 15 μM of {Cu} was used if not stated otherwise. This way, a strong PL quenching effect could be observed while keeping the {Cu} concentration as low as possible.

To investigate the origin of the quenching process, ethylenediaminetetraacetic acid (EDTA), a strong metal-chelating ligand, was added to the SWNT dispersion. The initially induced attenuation of PL intensity and shift of the PL peak position were immediately reversed and the optical properties of the functionalized (6,5) SWNTs were recovered (see Supplementary Fig. 2). Hence, we can assume that $Cu^{2+}$ ions do not lead to permanent but to reversible changes of the luminescent properties of SWNTs when present in their direct environment. The quenching mechanism will be discussed in more detail later. Importantly, this reversible quenching process is the basis for applying $sp^3$-functionalized (6,5) SWNTs as luminescent probes for biorelevant and strongly copper-complexing molecules such as $PP_i$. The corresponding detection scheme is outlined in Fig. 1a. Upon addition of {Cu}, the SWNT fluorescent probe goes into an off state and exhibits significantly reduced PL intensities as well as red-shifted peak positions. Identical to the observed effect with EDTA, the addition of $PP_i$ leads to a full recovery of the initial emission properties and the probe returns to its on state (see Fig. 1d).

### Quantitative detection of $PP_i$

To further explore the properties of functionalized SWNTs as near-infrared fluorescent probes, quantitative detection of $PP_i$ was tested.

 

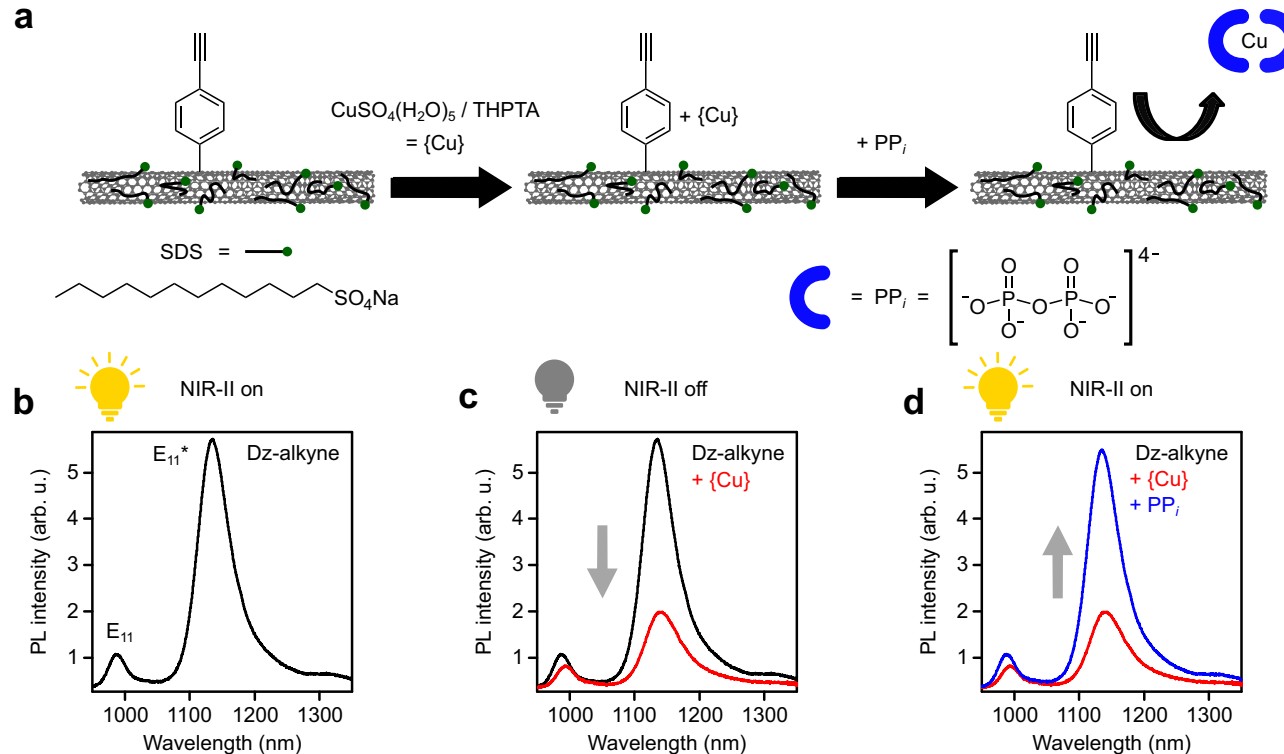

**Fig. 1 | Design strategy for detection of PP$_i$ with $sp^3$-functionalized (6,5) SWNTs.** **a** Dz-alkyne-functionalized SWNTs dispersed with SDS (ionic headgroups are indicated in green) display a high sensitivity towards the presence of a CuSO$_4$(H$_2$O)$_5$/THPTA complex ({Cu}); strong quenching of the E$_{11}$ (988 nm) and defect-induced E$_{11}$* (1135 nm) emission occurs. This effect is reversed upon addition of PP$_i$ (blue semicircle). **b** PL spectrum of Dz-alkyne. **c** PL spectra before (black) and after addition of {Cu} (red). **d** PL spectra before (red) and after addition of PP$_i$ (blue). The yellow lightbulb indicates bright E$_{11}$* emission at the corresponding step in the detection scheme. Source data are provided as a Source Data file.

15 µM of {Cu} were added to a dispersion of 4-ethynyl benzene functionalized (6,5) SWNTs and PL spectra were recorded after adjusting different concentrations of PP$_i$ and 15 min of incubation time (Fig. 2a−c). A calibration curve for the quantification of PP$_i$ in aqueous solution was obtained as the E$_{11}$ and E$_{11}$* emission gradually recovered for PP$_i$ concentrations between 12 µM–10.94 mM (see Fig. 2d). For higher concentrations of PP$_i$ (19.95 mM), the E$_{11}$ and E$_{11}$* emission decreased again. We assume that the drop in PL intensity at higher concentrations of PP$_i$ is caused by aggregation of SWNTs as it was also observed when no {Cu} was present (see Supplementary Fig. 3 and Supplementary Note 1). This does not affect the use of this sensor because these concentrations are much higher than in typical biological systems. Importantly, the E$_{11}$*/E$_{11}$ peak intensity ratio also increased for increasing PP$_i$ concentrations and remained stable even at high concentrations (see Fig. 2b, e). Thus, the peak intensity ratio is a more reliable detection metric as it is less influenced by aggregation effects. In addition to the peak intensity ratio, the PL area ratio also represents a suitable metric and shows nearly identical trends (see Supplementary Fig. 4a). The E$_{11}$* defect emission of functionalized SWNTs and its different sensitivity to their environment enable ratiometric detection, which is generally more reproducible and selective than absolute intensity measurements.

Finally, we tested the suitability of the E$_{11}$* (see Fig. 2c, f) and E$_{11}$ (Supplementary Fig. 4b, c) peak positions as quantitative metrics. Again, the induced red-shift upon addition of {Cu} continuously decreased and correlated directly with the concentration of added PP$_i$. All of these metrics showed a good correlation with the PP$_i$ concentration for a wide detection window from 100 µM–10 mM, thus confirming their capability to quantify the concentration of PP$_i$ in aqueous media with Dz-alkyne functionalized (6,5) SWNTs as NIR-II fluorescent probes.

## Selectivity of $sp^3$-functionalized SWNT probes

Due to the complex composition of biological environments, high selectivity is important for any sensor in addition to sensitivity. To explore the selectivity of functionalized (6,5) SWNTs as fluorescent probes for PP$_i$, they were quenched by 15 µM of {Cu} and PL spectra were recorded after the addition of 1 mM of potentially interfering analytes. Note that the concentrations of the tested molecules and anions are typically much lower in biologically relevant systems. Figure 3a, b shows the extracted PL intensity ratios and normalized E$_{11}$* intensities after the addition of CO$_3$$^{2-}$, NO$_3$$^-$, Cl$^-$, acetate (AcO$^-$), I$^-$, PO$_4$$^{3-}$, citrate, ADP, ATP, or PP$_i$ as well as L-cysteine. All normalized and absolute PL spectra including the extracted peak positions and optical trap depths are shown and listed in the Supplementary Information (Supplementary Fig. 5 and Supplementary Table 1).

The tested anions can be broadly categorized into weak copper-complexing and strong copper-complexing small molecules. For all analytes, for which weak coordination to Cu$^{2+}$ ions is expected (CO$_3$$^{2-}$, NO$_3$$^-$, Cl$^-$, AcO$^-$, I$^-$), only minor changes of the PL intensity ratio and the E$_{11}$* intensity were observed. In clear contrast to that, molecules that are known to strongly coordinate to Cu$^{2+}$ ions (PO$_4$$^{3-}$, citrate, ADP, ATP, PP$_i$, L-cysteine) showed a significant increase in the E$_{11}$*/E$_{11}$ PL intensity ratio. This trend was also valid for changes in the E$_{11}$* intensity, although much less pronounced. In all cases, the responses of the PL intensity ratio and E$_{11}$* intensity were still strongest for PP$_i$. Thus, while it may not be possible to detect low concentrations of PP$_i$ with high selectivity in the presence of other Cu$^{2+}$-complexing analytes, it should be possible to detect dynamic changes of the relevant concentration levels of PP$_i$.

The concentrations of PP$_i$ and ATP are closely correlated in living cells because PP$_i$ is a side-product of the hydrolysis of ATP. To investigate the possibility of tracking this hydrolysis reaction, we added

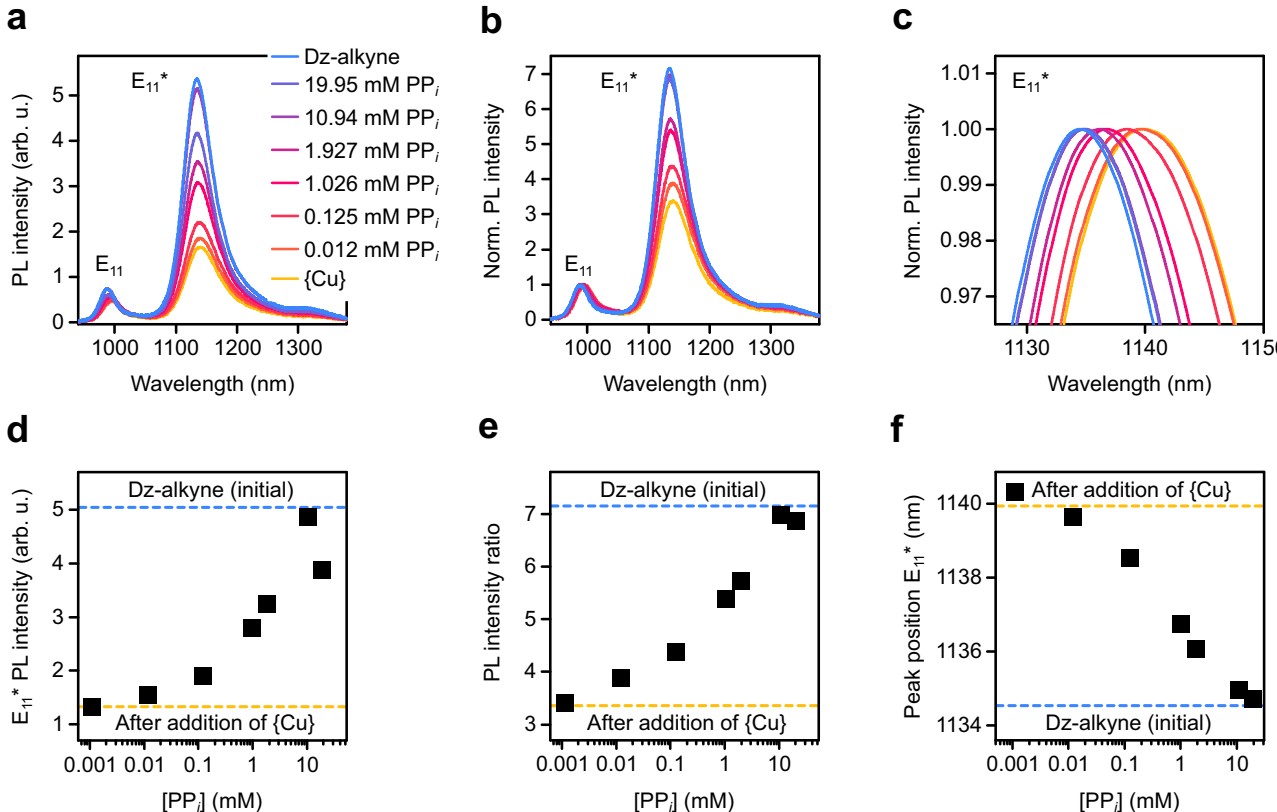

**Fig. 2 | Detection of PP$_i$ with $sp^3$-functionalized (6,5) SWNTs. a, b** Absolute PL spectra (**a**) and PL spectra normalized to E$_{11}$ (**b**) of Dz-alkyne (blue) alone and after the addition of {Cu} (15 μM, yellow) and various concentrations of PP$_i$. **c** Zoom-in on the normalized E$_{11}$* peak position. **d**–**f** E$_{11}$* PL intensity (**d**), E$_{11}$*/E$_{11}$ PL intensity ratio (**e**) and E$_{11}$* peak position (**f**) versus concentration of PP$_i$. The initial PL intensity, PL intensity ratio, and peak position of Dz-alkyne before and after the addition of {Cu} are indicated by blue and yellow dashed lines, respectively. Source data are provided as a Source Data file.

mixtures of ATP and PP$_i$ with a fixed overall concentration to a dispersion of functionalized (6,5) SWNTs that had been previously quenched with {Cu}. Figure 3c shows the recorded PL spectra before and after the addition of two different ATP:PP$_i$ mixtures (1:9 and 9:1) with a total concentration of 1 mM. While both ATP and PP$_i$ are expected to lead to a response by the SWNT sensor, a large concentration of PP$_i$ resulted in a significantly higher E$_{11}$*/E$_{11}$ PL intensity ratio than a large ATP concentration. It shows that there is selectivity, however, in mixed samples only the overall concentration of similar molecules can be measured. Similar results were observed for measurements with total analyte concentrations of 0.5 mM and 4.0 mM (see Supplementary Fig. 6). Furthermore, the E$_{11}$* peak position was redshifted by 2 nm for higher relative concentrations of PP$_i$. Note that this shift is opposite to the expected trend of an increasing blue-shift of the E$_{11}$* peak position for higher PL intensity ratios. We assume that the changes in E$_{11}$* peak positions are unique for each analyte and may offer additional parameters for identification.

In summary, $sp^3$-functionalized (6,5) SWNTs can be used as fluorescent probes to detect mixing ratios for ATP and PP$_i$. Hence, the hydrolysis of ATP could be followed in real-time by observing changes in the PL intensity ratios as well as shifts in the E$_{11}$* peak position. However, to develop more selective fluorescent probes, it is essential to understand the cause of PL quenching and recovery and the origin of the partial selectivity of $sp^3$-functionalized (6,5) SWNTs toward PP$_i$.

## PL quenching and sensing mechanism

We have demonstrated PL quenching of $sp^3$-functionalized (6,5) SWNTs by Cu$^{2+}$ and the resulting PP$_i$ detection capabilities. However, we have not yet discussed the role of the ligand THPTA or the attached 4-ethynylbenzene moiety for sensing. THPTA can be expected to have

a major impact on the PP$_i$ detection as it also forms complexes with Cu$^{2+}$ ions. The ligand itself has no effect on the emission of functionalized (6,5) SWNTs even at high concentrations (100 μM, see Supplementary Fig. 7). Thus, all PL changes (mainly quenching) can be attributed to the Cu$^{2+}$ ions. However, when Cu$^{2+}$ ions are added directly to functionalized SWNTs as a solution of CuSO$_4$(H$_2$O)$_5$ without THPTA, only moderate PL quenching was observed (see Fig. 4a). Much stronger quenching takes place when THPTA is also present. This enhancement can be explained with a higher effective concentration of Cu$^{2+}$ ions in close proximity to the SWNT surface when complexed by THPTA. The triazole THPTA strongly adsorbs to graphene and SWNTs via $\pi$-$\pi$ interactions[47,48]. A strong adsorption of the Cu$^{2+}$/THPTA complex (i.e., {Cu}) to the SWNT surface brings Cu$^{2+}$ ions in much closer and direct contact with the SWNTs.

The direct contact of Cu$^{2+}$ ions with the SWNT lattice or defect site further leads to red-shifts of the E$_{11}$ and E$_{11}$* emission. We presume these wavelength shifts to be caused by an increase in solvent polarity close to the SWNT surface or local changes in SWNT solvation due to the additional hydration shell around the adsorbed Cu$^{2+}$ ions. Similar solvatochromic shifts of SWNTs were described by Larsen et al. and Shiraki et al. for functionalized SWNTs[49,50].

Overall, stronger PL quenching and shifts of the emission peak positions can be observed for samples with THPTA compared to samples without it. When PP$_i$ is added, the Cu$^{2+}$/THPTA complex dissociates and a more stable Cu$^{2+}$/PP$_i$ complex is formed. The competitive complexation of the Cu$^{2+}$ ions by THPTA and PP$_i$ can be monitored by absorption spectroscopy due to the strong change in THPTA absorption upon complexation/decomplexation and further compared to changes in the E$_{11}$*/E$_{11}$ intensity ratio of the SWNT probe (see Supplementary Fig. 8 and Supplementary Note 2). The dissociation

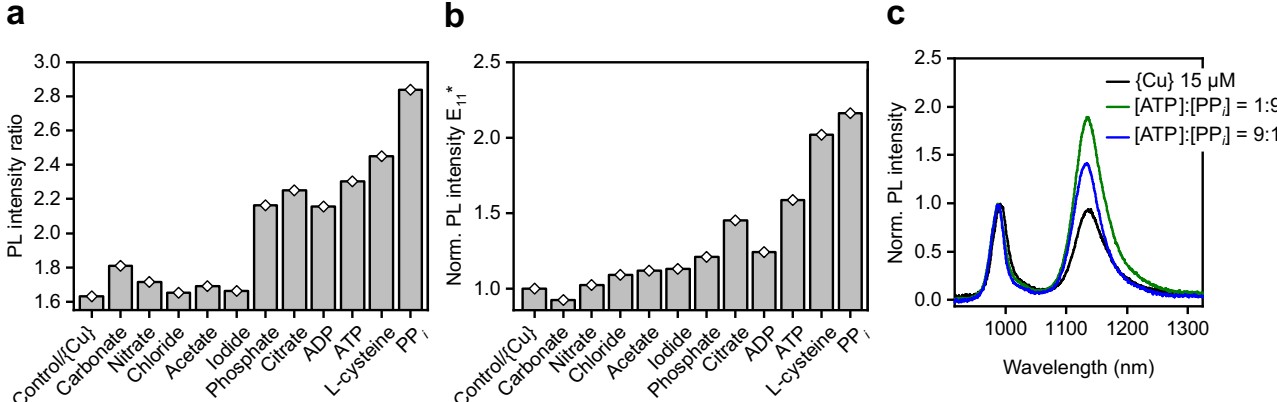

**Fig. 3 | Sensor response in the presence of other copper-complexing analytes.** **a**, **b** Response of $E_{11}^*/E_{11}$ PL intensity ratio (**a**) and normalized $E_{11}^*$ intensity (**b**) of Dz-alknye with {Cu} complex in the presence of various weakly or strongly copper-complexing analytes (concentration 1 mM, single measurements, $n = 1$). Note that the concentrations of the interfering analytes are much higher than one would expect in realistic analytical assays. **c** PL spectra after the addition of different ratios of ATP and $PP_i$ with a total analyte concentration of 1 mM. Source data are provided as a Source Data file.

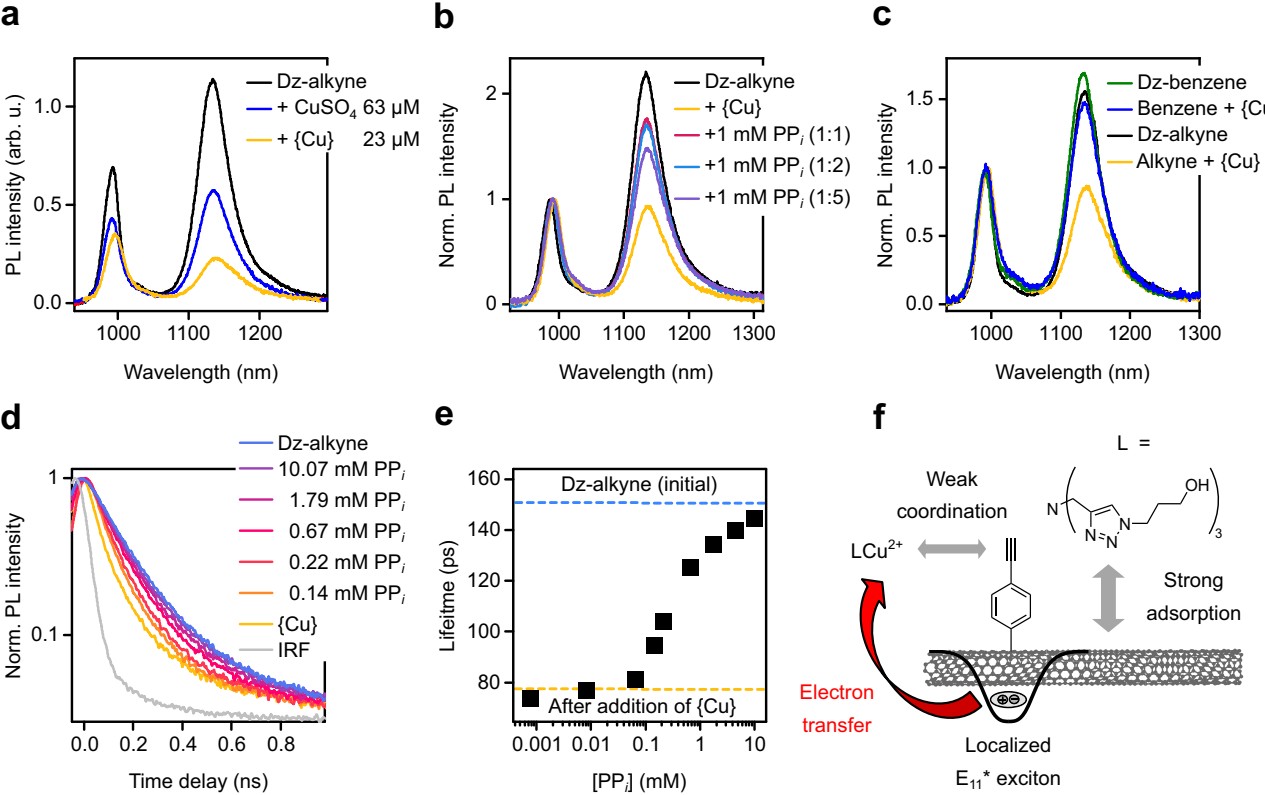

**Fig. 4 | Investigation of quenching mechanism. a** PL spectra of Dz-alkyne before and after the addition of 63 μM $CuSO_4(H_2O)_5$ or 23 μM $CuSO_4(H_2O)_5$/THPTA ({Cu}). **b** Normalized PL spectra before and after addition of 1 mM $PP_i$ for different concentrations of THPTA (15, 30, 75 μM) at a fixed concentration of $CuSO_4(H_2O)_5$ (15 μM) corresponding to ratios of 1:1, 1:2 and 1:5. **c** Normalized PL spectra of 4-ethynylbenzene and benzene-functionalized (6,5) SWNTs after the addition of {Cu} (15 μM). **d** TCSPC histogram of the PL decay after addition of different concentrations of $PP_i$. **e** Extracted amplitude averaged (Amp-avg.) lifetimes of $E_{11}^*$ emission from biexponential fits vs. concentration of $PP_i$. Initial values before and after addition of {Cu} are indicated by blue and yellow dashed lines, respectively. **f** Schematic presentation of the proposed adsorption and quenching mechanism. $Cu^{2+}$ ions are immobilized due to strong π-π interactions of the triazole ligand (L = THPTA) with the SWNT sidewall, which is further facilitated by weak coordination to the ethynyl groups. After exciton formation and diffusion to defect sites, $E_{11}^*$ excitons are quenched by fast electron transfer (red arrow) to the metal center.

constant for the $Cu^{2+}$/THPTA complex on the SWNT surface is 25 times higher than that of the free complex, indicating that the $Cu^{2+}$/$PP_i$ complex formation is weakened due to steric shielding of the $Cu^{2+}$ center by adsorption to the SWNT surface.

Based on this concept, the selectivity of the fluorescent SWNT probes toward different analytes (see above) can be rationalized by competitive complexation of $Cu^{2+}$ ions by THPTA and the analyte. Upon complexation of $Cu^{2+}$ ions by an analyte, they are removed from

the direct environment of the SWNT and the $sp^3$ defect. Thus, the original PL is recovered. When $PP_i$ (1 mM) sensing is performed at higher THPTA concentrations, the response is markedly lower as the complexation equilibrium is shifted towards the $Cu^{2+}$/THPTA complex formation (see Fig. 4b). Consequently, sensitivity and selectivity should be tunable by ligand choice and concentration, creating perspectives for tailoring SWNT fluorescent probes with higher selectivity towards specific analytes. Future work should focus on the design of ligand systems that show a strong physisorption to the SWNT surface and suitable complexation strength with $Cu^{2+}$ compared to, for example, $PP_i$. The $Cu^{2+}$/triazole complex is a good starting point as it shows strong adsorption, good selectivity toward phosphates and was previously used to design molecular sensors[51].

As the adsorption of $Cu^{2+}$ ions on the SWNT surface clearly plays a key role in the PL quenching process, (6,5) SWNTs may also be $sp^3$-functionalized with other moieties such as simple benzene groups instead of 4-ethynylbenzene. However, PL quenching was found to be significantly stronger for 4-ethynylbenzene-functionalized SWNTs at lower concentrations of {Cu} (see Fig. 4c and Supplementary Fig. 9). Presumably the additional weak coordination of copper ions to alkyne groups further enhances their interaction with the SWNT. This notion is corroborated by the observed quenching behavior of the $E_{11}$ emission, which is similar for pristine and benzene-functionalized SWNTs, but enhanced for 4-ethynylbenzene-functionalized SWNTs (see Supplementary Fig. 10 and Supplementary Note 3). This dependence on functionalization highlights that while the presented detection scheme could be transferred to (6,5) SWNTs with different functional groups, the effect is stronger with the ethynyl moiety. However, the type of alkyne moiety can be varied. For instance, the functionalization of (6,5) SWNT with 2-ethynylbenzene, leads to another further redshifted defect-emission band located around 1242 nm as previously shown by Yu et al.[52]. Nanotubes that were functionalized in this way also showed {Cu} and $PP_i$ responsiveness but with emission even further in the NIR-II (see Supplementary Fig. 11 and Supplementary Table 2).

The adsorption of $Cu^{2+}$ on the SWNT surface evidently causes PL quenching, which is recovered when analytes are added that form stable complexes with $Cu^{2+}$, however, the underlying cause for the initial quenching is still an open question. UV-Vis-NIR absorption spectra of functionalized (6,5) SWNTs recorded before and after the addition of {Cu} reveal only a 5 nm red-shift of the $E_{11}$ transition (see Supplementary Fig. 12) and no bleaching. While aggregation or bundling of SWNTs would lead to a similar shift, we can exclude this explanation due to the full reversibility of the quenching process.

To investigate the possibility of the formation of non-emissive SWNT-$Cu^{2+}$ ground-state complexes, PL decays of the $E_{11}^*$ defect-state were recorded by time-correlated single-photon counting (TCSPC) after the addition of {Cu} and at different concentrations of $PP_i$ (see Fig. 4d). Note that the fast $E_{11}$ PL decay could not be resolved (see instrument response function, IRF) and will not be considered here. The $E_{11}^*$ defect-state emission typically shows a biexponential decay. The short and long lifetime components can be averaged according to the weights of their normalized amplitudes[53]. The obtained amplitude-averaged lifetimes ($\tau_{amp-avg.}$) depending on the added {Cu} followed by $PP_i$ can be used to distinguish between quenching due to additional non-radiative decay paths or ground-state quenching.

Upon addition of {Cu} the amplitude-averaged $E_{11}^*$ lifetime decreases from 151 to 79 ps by a factor of 1.9, which is in good agreement with the 2.3-fold reduction of the integrated $E_{11}^*$/$E_{11}$ intensity ratio (see Table 1). This correlation of lifetime and quenching factor remains valid upon the addition of $PP_i$ and the initial $E_{11}^*$ lifetime is restored for high concentrations of $PP_i$ (see Fig. 4e; for full data set see Supplementary Fig. 13 and Supplementary Table 3). Hence, the reduction of the $E_{11}^*$ emission should be mainly due to additional non-radiative decay paths and not connected to the formation of a non-emissive ground state complex, which would have no impact on the PL

## Table 1 | Extracted PL lifetimes and quenching factors

| Sample | $\tau_{amp-avg.}$ (ps) | QF($\tau_{amp-avg.}$) | Area ratio | QF |
|---|---|---|---|---|
| Dz-alkyne | 151 | – | 3.47 | – |
| {Cu} | 79 | 1.91 | 1.54 | 2.25 |
| 0.67 mM $PP_i$ | 125 | 1.21 | 2.48 | 1.40 |
| 10.07 mM $PP_i$ | 145 | 1.04 | 3.27 | 1.06 |

Extracted $\tau_{amp-avg.}$ and $E_{11}$/$E_{11}^*$ PL area ratios with corresponding quenching factors (QF). For full data set see Supplementary Table 3.

lifetime. The robust correlation of the $E_{11}^*$ lifetime and $PP_i$ concentration may even enable the implementation of functionalized (6,5) SWNTs as probes for next-generation fluorescence-lifetime imaging microscopy in the near-infrared[54].

Quenching of fluorophores by $Cu^{2+}$ ions is most commonly attributed to either their paramagnetic properties or fast electron transfer to the metal center. Paramagnetic effects are unlikely to cause the observed quenching as no correlation with the magnetic moment was found for PL quenching with other paramagnetic ions such as $Ni^{2+}$ and $Co^{2+}$ (see Supplementary Fig. 14). This absence of paramagnetic effects is in agreement with previous studies by Brege et al.[55]. Consequently, we suggest that fast photoinduced electron transfer (PET) is the primary cause for the observed PL quenching. This attribution is supported by the estimated Gibbs free energy for PET in this system, confirming that electron transfer is thermodynamically favorable (see Supplementary Note 4)[56].

Figure 4f provides an overview of the proposed $Cu^{2+}$ adsorption and PL quenching mechanism. The stronger quenching effect of $Cu^{2+}$ ions on the $E_{11}^*$ emission can be rationalized by the coordination of {Cu} to the ethynyl groups close to the $sp^3$ defects and the longer lifetime of defect-localized excitons and hence more likely PET compared to $E_{11}$ excitons with only ps lifetimes. Further studies of the influence of the defect density of functionalized SWNTs on the sensitivity toward {Cu} and $PP_i$ revealed that while the quenching factor of the $E_{11}^*$/$E_{11}$ PL intensity ratio is largely unaffected by the $sp^3$ defect density, the highest sensitivity for $PP_i$ detection is obtained at low defect densities (see Supplementary Fig. 15 and Supplementary Note 5 for further discussion). These insights into the underlying PL quenching by $Cu^{2+}$ ions and $PP_i$ sensing mechanism by $sp^3$-functionalized SWNTs can be used to further optimize detection schemes for specific analytes by metal-displacement fluorescent probes.

## Detection of $PP_i$ in biological environments

All of the previous experiments were performed with SDS-dispersed (6,5) SWNTs, which are not biocompatible due to the required excess surfactant. To achieve biocompatibility and enable $PP_i$ detection in biological environments, the SDS surfactant of the functionalized (6,5) SWNTs was replaced with phospholipid-polyethylene glycol (PL-$PEG_{5000}$) as previously reported by Welsher et al.[8] (for details see Methods). Successful surfactant exchange was confirmed by absorption and PL spectroscopy. In agreement with previous reports[14], a 6 nm red-shift of the $E_{11}$ transition was observed while all other spectroscopic features remained the same (see Supplementary Fig. 16). Again, {Cu} was added to PL-$PEG_{5000}$-coated and $sp^3$-functionalized (6,5) SWNTs. After 15 min of incubation the dispersion was spin-filtered (cut-off of 100 kg $mol^{-1}$) and re-dispersed in 10 mM EDTA-free MOPS buffer. Excess {Cu} was removed by the filtration step while physisorbed {Cu} was expected to remain on the SWNT surface. After dilution to a concentration corresponding to an absorbance of 0.1 at the $E_{11}$ transition (1 cm path length), PL spectra were recorded and are shown in Fig. 5a, b. The total PL quenching was reduced compared to SDS-dispersed SWNTs. This change might be attributed to the different coverage of the SWNT sidewalls by PL-$PEG_{5000}$ compared to SDS. The polymeric PL-$PEG_{5000}$ was previously estimated to cover the SWNT

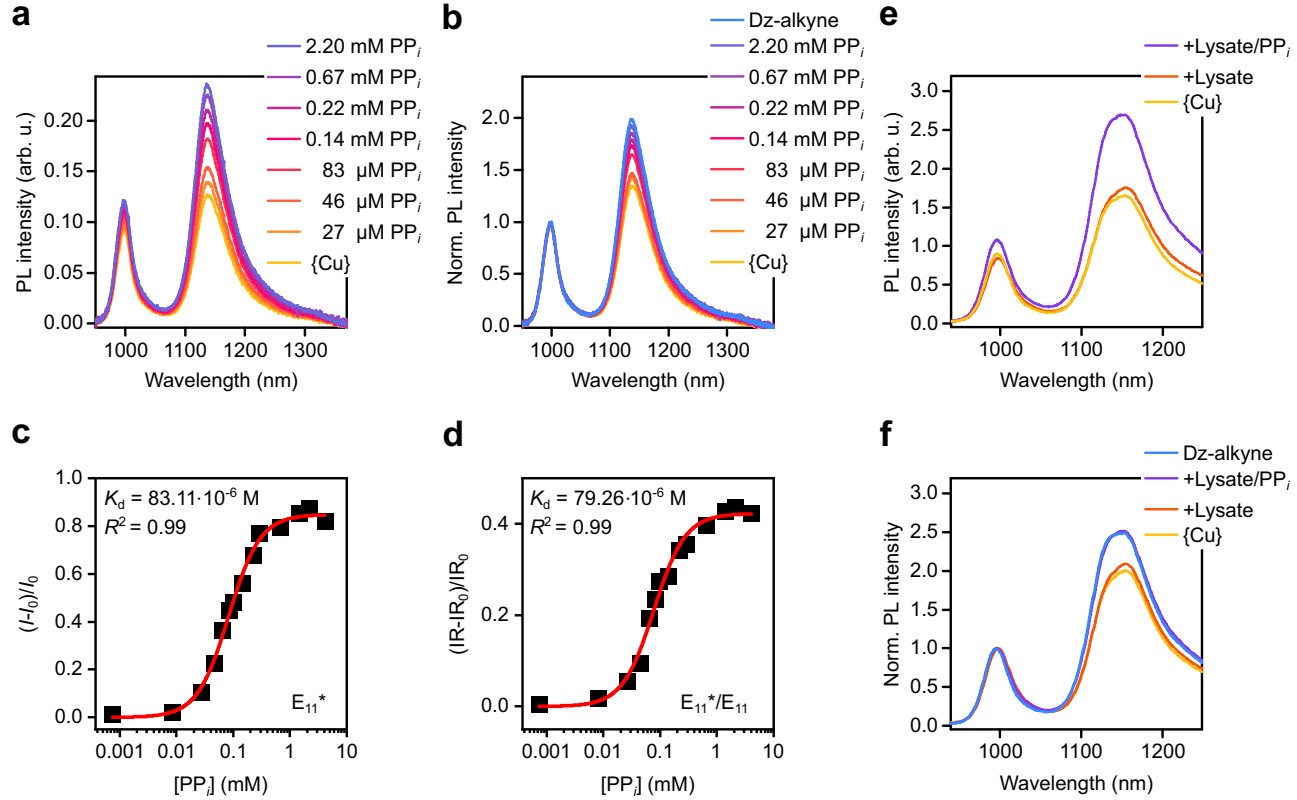

**Fig. 5 | Detection of PP_i in surfactant-free biological buffer. a, b** Absolute (**a**) and normalized (**b**) PL spectra of Dz-alkyne-functionalized SWNTs dispersed in PL-PEG_5000 and 10 mM MOPS buffer after addition of {Cu} and various concentrations of PP_i. **c** PP_i concentration-dependent changes of the $E_{11}^*$ PL intensity with $I$ as the $E_{11}^*$ intensity and $I_0$ as $E_{11}^*$ intensity after addition of {Cu} including Hill function fit to the data (red line). **d** PP_i concentration-dependent changes of the $E_{11}^*/E_{11}$ PL intensity ratio with IR as the $E_{11}^*/E_{11}$ intensity ratio and $IR_0$ being the $E_{11}^*/E_{11}$ intensity ratio after addition of {Cu} including Hill function fit to the data (red line). The coefficient of determination ($R^2$) and $K_d$ values of the fits are given in the respective plots. **e, f** Absolute (**e**) and normalized (**f**) PL spectra before and after the addition of {Cu} and addition of 0.5 μL cell-lysate, which was subsequently spiked with 188 μM PP_i. PL spectra in (**e**) and (**f**) were recorded at lower SWNT concentrations resulting in higher sensitivity upon addition of PP_i. Wavelength scales are different due to the use of a different spectrometer and diffraction grating (see Methods). Additional peak broadening of the $E_{11}^*$ emission may originate from functionalization at a higher diazonium salt concentration, yielding more red-shifted emission bands[69]. Source data are provided as a Source Data file.

surface more completely than short SDS molecules, which should reduce interaction with and hence quenching by adsorbed {Cu}[12,57]. A similar effect can be observed for DOC-coated and functionalized (6,5) SWNTs, for which almost no PL quenching occurs after the addition of $Cu^{2+}$ ions, most likely due to the dense packing of DOC on the surface (see Supplementary Fig. 17)[58].

While the response toward $Cu^{2+}$ ions was reduced by PL-PEG_5000, the sensor remained fully operational and showed good sensitivity towards the presence of PP_i. Furthermore, the PL-PEG_5000-dispersed SWNTs showed no aggregation effects even at high concentrations of PP_i (see Supplementary Fig. 18). Figure 5c, d displays the concentration-dependent PL response of the $E_{11}^*$ emission and corresponding PL intensity ratios fitted with a Hill function[59]:

$$Y = \frac{[PP_i]^n}{K_d^n + [PP_i]^n} \qquad (1)$$

where [PP_i] is the PP_i concentration, $K_d$ is the dissociation constant and $n$ the Hill coefficient (for response of $E_{11}$ emission, PL area ratios and peak positions see Supplementary Fig. 19).

To confirm the reproducibility of this sensor, we performed additional concentration-dependent PL measurements using (6,5) SWNTs functionalized with a higher $sp^3$-defect density (see Supplementary Figs. 20, 21). Again, a similar concentration-dependent fluorescence response can be observed and the absolute PL intensity and PL intensity ratios appear to be equally suited to detect PP_i. However, the concentration-dependent absolute PL intensities and peak positions of both batches of functionalized (6,5) SWNTs display significant deviation. In contrast to that, nearly identical response curves are obtained when using the ratiometric approach by comparing the $E_{11}^*/E_{11}$ PL intensity ratios (see Supplementary Fig. 22).

The PL intensity ratios are clearly superior to absolute values and confirm that the developed ratiometric fluorescent PP_i sensor is robust and reproducible even between different nanotube batches with different degrees of functionalization. Corresponding $K_d$ values and Hill parameters for both defect densities can be found in Supplementary Table 4. We find a usable detection range from 10% to 90% of the maximum response of the PL intensity ratios of the biocompatible SWNT-probes in buffer for a PP_i concentration from 17 to 320 μM. This range depends to some degree on the SWNT concentration and could be adjusted with respect to signal-to-noise ratio and application.

The limit of detection (LOD) was determined to be 4.4 μM for the $E_{11}^*/E_{11}$ PL intensity ratios (5.2 μM and 12.5 μM for $E_{11}$ and $E_{11}^*$ intensity, respectively, see Supplementary Fig. 23 and Supplementary Note 6). Many clinical conditions that cause increased levels of PP_i in urine, intracellular mitochondrial or synovial fluids[60–66] could be traced within this detection range. For example, patients with hypophosphatasia show increased levels of PP_i with concentrations between 5 μM and 17.5 μM in plasma and 45–234 μM in urine[60–63].

As for most fluorescent sensors, calibration curves must be obtained to correctly quantify PP_i concentrations based on the $E_{11}^*/E_{11}$ intensity ratios for specific biological environments. Metrics that are

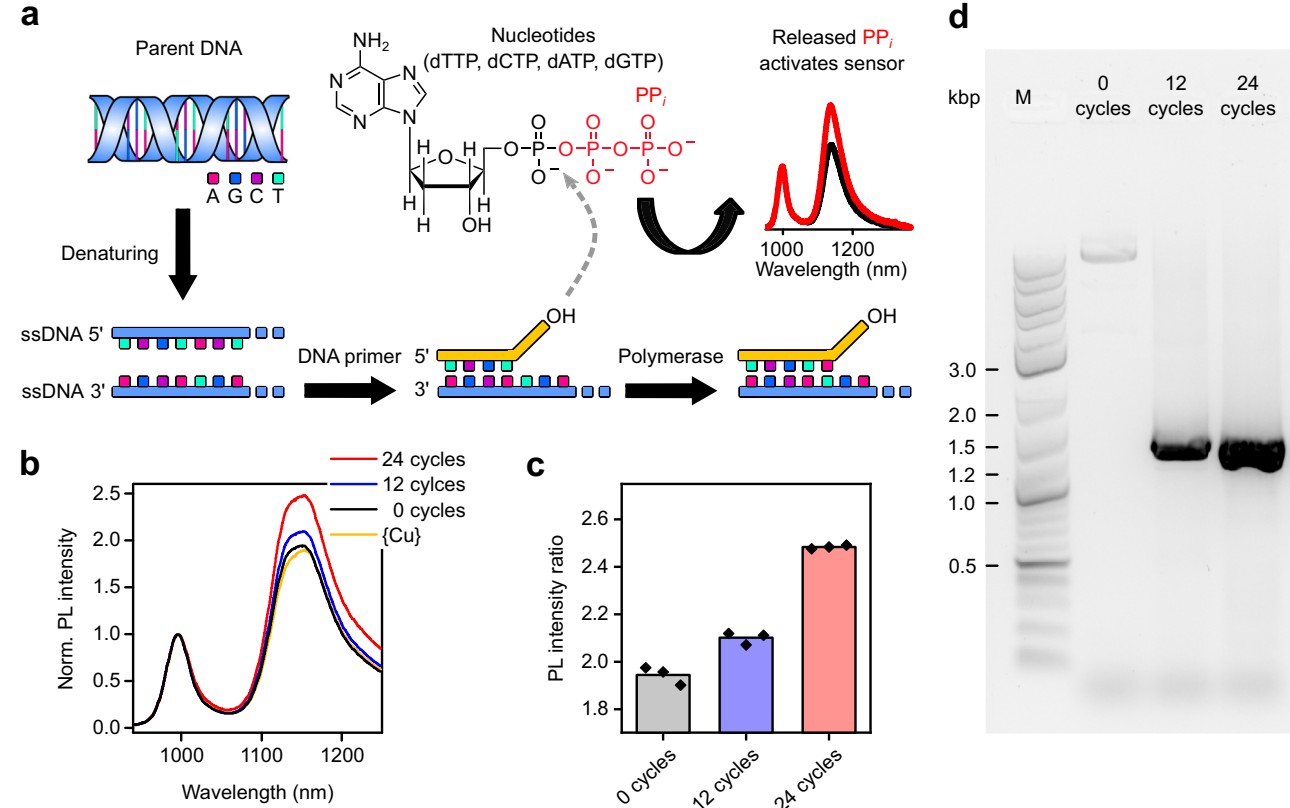

**Fig. 6 | Monitoring PCR cycles by detecting released PP$_i$. a** Concept for measuring PCR cycles by detecting the released PP$_i$ with the SWNT-sensor. The DNA backbone is presented in blue, the DNA primer is presented in yellow. Nucleobases are shown as magenta (A, adenine), dark blue (G, guanine), purple (C, cytosine) and turquoise (T, thymine) squares. Released PP$_i$ originating from dNPTs (here dATP) is shown in red. **b** Normalized and averaged (3 measurements) PL spectra of Dz-alkyne dispersed with PL-PEG$_{5000}$ and treated with {Cu} after addition of 5 μL of PCR product after 0, 12, and 24 cycles. **c** Measured PL E$_{11}$*/E$_{11}$ intensity ratios after

the addition of 5 μL of PCR product after 0, 12, and 24 cycles. One biologically independent sample was examined over $n = 3$ independent measurements with the SWNT sensor, and should represent the spread in the response of the sensor. The height of the bar represents the average of the three measurements. **d** Gel-electrophoresis of PCR products after 0, 12, and 24 cycles, M−DNA marker with length of DNA fragments in kilo-base-pair (kbp). Source data are provided as a Source Data file.

more sensitive toward the dielectric environment of SWNTs such as absolute PL intensities and PL peak positions will require even more advanced adjustments. A previously reported NIR-II fluorescent sensor for PP$_i$ based on lanthanide nanoparticles showed similar sensitivity and higher selectivity toward PP$_i$ but lacked the advantage of internal calibration and multimodal sensing enabled by Dz-alkyne functionalized SWNTs[39].

To explore the potential of this fluorescent PP$_i$ sensor in a biological environment, we cultured HEK cells and carefully performed a washing step to remove the phosphate-containing growth medium before the cells were lysed. Upon addition of cell lysate to the {Cu}-quenched SWNT sensor and subsequent spiking with PP$_i$, a clear increase in PL intensity (see Fig. 5e), a change in the PL intensity ratio (see Fig. 5f) and PL peak position (see Supplementary Fig. 24) occurred. It should be noted, that under the given conditions a small increase in E$_{11}$* intensity and E$_{11}$*/E$_{11}$ intensity ratio were already observed for unspiked lysate. This offset was expected, because significant amounts of phosphate and PP$_i$ were already present for the chosen lysate concentration[67]. Overall, PL-PEG$_{5000}$-coated and $sp^3$-functionalized SWNTs can be used as NIR-II fluorescent PP$_i$ probes under biological conditions and in complex media. Hence, they should be suitable as a bioanalytical tool.

### Detection of PP$_i$ released from PCR

Finally, the developed SWNT sensor was used to measure the PP$_i$ that is released during DNA strand synthesis by PCR to enable online

monitoring. In a PCR (see Fig. 6a), exponential duplication of a template DNA is achieved and PP$_i$ is released from deoxynucleoside triphosphates (dNPTs) when DNA synthesis takes place by DNA polymerase. To track the released PP$_i$ during the amplification process, it is important to detect small amounts of PP$_i$ in the presence of structurally similar dNPTs, DNA polymerase enzyme, DNA template (plasmid), and DNA oligos, which are used as primers and PCR buffer.

To test the PP$_i$ sensor in such a complex system, PCR was performed with 0, 12, and 24 cycles and 5 μL of the final PCR product was added to a {Cu}-quenched PL-PEG$_{5000}$-coated and $sp^3$-functionalized (6,5) SWNT dispersion. As expected, a slight response was already observed for zero cycles due to the high concentration of dNPTs. A different response was evident after the addition of PCR product, when PCR was performed for 12 and 24 cycles (see Fig. 6b). An increase of the E$_{11}$*/E$_{11}$ PL intensity ratio occurred immediately and no incubation time was required.

It should be noted that in the presented case the conversion of dNPTs to PP$_i$ is detected by the SWNT sensor not the absolute PP$_i$ concentration. This type of response has several implications. Compared to the previous data, where PL spectra were collected upon the addition of PP$_i$, the expected shifts in peak position have changed. This effect was already apparent for different mixing ratios of PP$_i$ and ATP (see Fig. 3 and Supplementary Fig. 6). Due to the structural similarity of dNPTs to ATP we expect a similar behavior. The conversion of dNPTs to PP$_i$ results in a red-shift of the E$_{11}$ and E$_{11}$* emission in contrast to the usually observed blue-shift (see Supplementary Fig. 25). Moreover, the

sensitivity range of the SWNT sensor is expected to increase. The conversion of dNPTs to $PP_i$ leads to a lower response in comparison to the direct addition of $PP_i$, thus allowing the sensor to track changes in $PP_i$ concentration over a larger concentration window. Nevertheless, the response resulting from the dNPTs to $PP_i$ conversion should still scale logarithmically. As the release of $PP_i$ by PCR and the response of the sensor both increase on a logarithmic scale, a linear correlation between the PL intensity ratio and the number of PCR cycles (12 and 24) is expected and is indeed observed (see Fig. 6c). A similar trend can be found for the absolute $E_{11}^*$ intensity and $E_{11}^*/ E_{11}$ PL area ratio (see Supplementary Fig. 25). Gel-electrophoresis revealed the successful amplification of the parent DNA of the same molecular weight (see Fig. 6d) and the band intensity of the visualized DNA products matches the observed trend for the PL intensity ratios. In summary, the developed fluorescent SWNT-sensor for $PP_i$ could be used as an alternative and instant probe to track DNA amplification during PCR in real-time over a large detection range in contrast to other time-consuming detection schemes such as gel-electrophoresis.

## Discussion

In summary, we have designed and presented $sp^3$-functionalized (6,5) SWNTs with well-defined emission features in the NIR-II for the ratiometric optical detection of $PP_i$ in biological environments. The $PP_i$ sensing scheme relies on the recovery of the near-infrared fluorescence of the nanotubes by copper ion displacement. The emission of covalently functionalized (6,5) SWNTs is quenched through photo-induced electron transfer to $Cu^{2+}$ ions that are immobilized on the SWNT surface by a triazole ligand and additional weak coordination to the covalently attached alkyne moieties. PL quenching is more pronounced for the $sp^3$ defect-related $E_{11}^*$ (1135 nm) emission than for the $E_{11}$ (988 nm) emission and leads to corresponding changes in the $E_{11}^*/ E_{11}$ intensity ratio in addition to the peak positions. The removal of $Cu^{2+}$ from the SWNT surface by $PP_i$ results in a restoration of the emission properties depending on the absolute concentration of $PP_i$, which enables a robust ratiometric detection scheme. While other strongly copper-complexing analytes (e.g., ATP) also cause a fluorescence response, the relative differences can be used to track biologically relevant processes. In comparison to other multi-signal sensors based on mixed chiralities of SWNTs, the emission features of monochiral (6,5) SWNTs are narrow, occur from the same species and thus offers a more reliable optical readout. The observed changes in relative and absolute PL intensities including analyte-dependent shifts in the $E_{11}$ and $E_{11}^*$ peak positions may also be used for detection schemes based on machine-learning algorithms.

Mechanistic studies indicate that the efficiency of PL quenching by $Cu^{2+}$ depends on the type of functional moiety attached to the $sp^3$ defects (here an aryl alkyne) and the ligand system (here a triazole with strong $\pi$-$\pi$ interactions) that coordinate the cupric ions close to the SWNT surface and enable fast photoinduced electron transfer. These insights will facilitate the design of ligand systems and metal-ion immobilization strategies for the development of optical sensors based on $sp^3$-functionalized SWNTs.

The current sensor design causes an irreversible change in fluorescence in the presence of $PP_i$, which is a universal feature of metal-displacement assays. In this detection scheme reversibility might only be achievable by the addition of new $Cu^{2+}$ ions or a new immobilization strategy where $Cu^{2+}$ complexation by $PP_i$ only impacts its distance to the SWNT surface and thus degree of PL quenching. Yet, irreversibility represents an advantage in assays that are supposed to provide robust and quantitative information at specific time points.

Importantly, the demonstrated SWNT fluorescent sensor remains fully operational with a detection window over two orders of magnitude when made biocompatible by coating with phospholipid-polyethylene glycol. It can be applied for the detection of $PP_i$ in lysate with high cell count and for the instant fluorescent detection of

$PP_i$ released during DNA synthesis in PCR with potential applications in PCR quality control. Due to its internal calibration and multiple sensing parameters, this detection scheme greatly expands the currently available methods for the detection of biomarkers in the NIR-II and opens the path towards in vivo detection of $PP_i$ with low background noise. Overall, the covalent functionalization of SWNTs with $sp^3$ defects provides additional fluorescence features and specific functional groups for selective and reliable optical (bio)sensing.

## Methods

### Materials

The following reagents were purchased from Sigma-Aldrich: nickel(II) sulfate hexahydrate (≥98%), cobalt(II) sulfate heptahydrate (≥99%), adenosine 5′-diphosphat sodium salt (ADP, ≥95%, bacterial, HPLC), sodium carbonate (≥99.5%), potassium acetate (98%), potassium nitrate (99%), potassium iodide (99%), calcium dichloride hexahydrate (98%), sodium citrate tribasic dihydrate, L-cysteine (97%), ethylene-diaminetetraacetic acid tetrasodium salt hydrate (EDTA), sodium pyrophosphate (98%), DOC (BioXtra, ≥98.0%), SDS (≥99.0%), sodium cholate (SC, ≥99.0%), sodium hypochlorite (NaClO, 10–15% active chlorine), tetrafluoroboronic acid (48 wt.%), tert-butyl nitrite (90%), 4-ethynylaniline (97%), 2-ethynylaniline (98%), aniline (≥99.5%), $CuSO_4(H_2O)_5$ (≥99.9%). The following reagents were purchased from TCI: adenosine 5′-triphosphate disodium hydrate (ATP, >98%, HPLC), dextran ($M_w$ = 70 kg mol⁻¹), THPTA (>97%). Poly(ethylene glycol) (PEG, $M_w$ = 6,000 g mol⁻¹) was purchased from Alfa Aesar. PL-$PEG_{5000}$ (18:0 PEG5000PE, 1,2-distearoyl-sn-glycero-3-phosphoethanolamine-N-[methoxy(polyethylene glycol)−5000], $M_w$ = 5801.071 g mol⁻¹) was purchased from Avanti Polar Lipids, Inc.

### Preparation of (6,5) SWNT dispersions

Dispersions of monochiral (6,5) SWNTs were prepared from CoMoCAT raw material (CHASM SG65i-L58) by ATPE[44]. ATPE was performed in a two-phase system composed of dextran and PEG. SWNTs were separated in a diameter sorting protocol with DOC and SDS. At a fixed DOC concentration of 0.04% (w/v), the SDS concentration was increased to 1.1% (w/v) to push all species with diameters larger than (6,5) SWNTs into the top phase for extraction. Then the SDS concentration was increased from 1.2% to 1.5% and all (6,5) SWNT enriched phases were collected. Separation of metallic and semiconducting SWNTs was achieved by further addition of SC and NaOCl as an oxidant. The selected (6,5) SWNTs were concentrated in a pressurized ultrafiltration stirred cell (Millipore) with a $M_w$ = 300 kg mol⁻¹ cut-off membrane and adjusted to 1% (w/v) SDS for further functionalization.

### Characterization methods

Absorption spectra with baseline correction were acquired with a Cary 6000i UV-VIS-NIR spectrophotometer (Varian, Inc.). PL spectra were measured at low excitation densities at the $E_{22}$ transition either with the unfocused wavelength-filtered output of a ps-pulsed super-continuum laser (NKT Photonics SuperK Extreme) or a 450 W Xe arc lamp and recorded using an Acton SpectraPro SP2358 spectrograph (grating blaze 1200 nm, 150 lines mm⁻¹) and a liquid nitrogen-cooled InGaAs line camera (Princeton Instruments, OMA-V:1024) or a Fluor-olog spectrofluorometer (HORIBA) equipped with a liquid nitrogen-cooled InGaAs line camera. For spiking and PCR experiments, PL spectra were measured under excitation with a 561 nm laser at 100 mW (Gem 561, Laser Quantum) and recorded with 4 s integration time using a spectrometer (Shamrock 193i, Andor Technology Ltd.) coupled to a microscope (IX73, Olympus). PL lifetimes of luminescent defect states were measured and analyzed in a time-correlated single photon counting scheme[19]. Briefly, functionalized (6,5) SWNTs were excited at the $E_{22}$ transition with a ps-pulsed supercontinuum laser (NKT Photonics SuperK Extreme) and the spectrally filtered emission was focused onto a gated InGaAs/InP avalanche photodiode (Micro Photon

Devices) with read-out by a PicoHarp 300 photon counting module (PicoQuant). Raman spectra were recorded with a Renishaw inVia Reflex confocal Raman microscope. Dispersions of SWNTs were drop-cast on glass substrates and rinsed carefully with ultra-pure water. A 532 nm laser was used for excitation and more than 1000 spectra were collected and averaged. Spectra were manually baseline-corrected by fitting a smooth cubic spline curve through points where only back-ground noise was expected.

## Synthesis of arenediazonium tetrafluoroborates

Ethynyl benzene and benzene diazonium salts were synthesized from the corresponding anilines (see also Supplementary Fig. 26)[68]. In a 25 mL flask, the aniline (0.85 mmol) was dissolved in acetonitrile (2 mL) and an aqueous solution of tetrafluoroboronic acid (222 μL, 149 mg, 2.0 eq.) was added. The solution was cooled to 0 °C in an ice/water bath and *tert*-butyl nitrite (223 μL, 193 mg, 2.2 eq.) was added dropwise. The mixture was stirred at 0 °C for 30 min and diethyl ether (10 mL) was added to precipitate the arenediazonium tetrafluoroborate. The obtained solid was filtered off, washed with cold diethylether (3 × 10 mL) and recrystallized from acetone. The arenediazonium tetra-fluoroborate was dried *in vacuo* for 1 h. 4-ethynylbenzene diazonium tetrafluoroborate was synthesized starting from 4-ethynylaniline. The diazonium product was recovered as a crystalline colorless powder (120 mg, 65%). 2-ethynylbenzene diazonium tetrafluoroborate was synthesized starting from 2-ethynylaniline. The diazonium product was recovered as a crystalline colorless powder (110 mg, 60%). Benzene diazonium tetrafluoroborate was synthesized starting from ani-line. The diazonium product was recovered as a crystalline colorless powder (141 mg, 86 %). Successful synthesis was confirmed with $^1H$ nuclear magnetic resonance spectroscopy (NMR, see Supplementary Note 7). The obtained diazonium salts were stored at −20 °C.

## $sp^3$-functionalization protocol

For functionalization of (6,5) SWNTs, the optical density of the aqu-eous dispersion was adjusted to 0.33 cm$^{-1}$ at the $E_{11}$ transition with ultra-pure water. Stock solutions of the corresponding diazonium salts with a final concentration of 10 μg mL$^{-1}$ were prepared and aliquots were added to the dispersion. Typically, reaction volumes of 315 mL and diazonium salt concentrations between 0.025 and 0.005 μg mL$^{-1}$ were used. For the functionalization with 2-ethynyl benzene diazonium tetrafluoroborate a final concentration of 0.06 μg mL$^{-1}$ of diazonium salt was used. All reaction mixtures were stored in the dark for 7 days. For pyrophosphate sensing with functionalized (6,5) SWNTs with SDS surfactant, excess diazonium salt was removed via multiple spin-filtration steps (Amicon Ultra-4, $M_w$ = 100 kg mol$^{-1}$) and functionalized (6,5) SWNTs were resuspended in 0.33% (w/v) SDS. All dispersions were sonicated for 15 min before further characterization.

## Pyrophosphate sensing protocol

For sensing of pyrophosphates in dispersions of functionalized (6,5) SWNTs with SDS surfactant, fresh stock solutions in ultra-pure water of CuSO$_4$(H$_2$O)$_5$ (12 mM) and THPTA (12 mM) were prepared and com-bined in a 1:1 ratio. Typically, 2.5 μL (corresponding to a final con-centration of the copper-complex of 15 mM) were added to 1 mL of $sp^3$-functionalized (6,5) SWNT dispersion with an optical density of 0.1 cm$^{-1}$ at the $E_{11}$ transition and incubated for 15 min.

For sensing of pyrophosphates with biocompatible SWNTs, dispersions of $sp^3$-functionalized (6,5) SWNTs were mixed with PL-PEG$_{5000}$ such that the final concentration of PL-PEG$_{5000}$ was 2 mg mL$^{-1}$. The mixture was transferred to a 1 kg mol$^{-1}$ dialysis bag (Spectra/Por®, Spectrum Laboratories Inc.) and dialyzed for 7 days against ultra-pure water to remove SDS. The obtained dispersion was sonicated for 15 min yielding PL-PEG$_{5000}$ dispersed $sp^3$-func-tionalized (6,5) SWNTs. 2 mL of dispersion were concentrated by spin-filtration (Amicon Ultra-4, $M_w$ = 100 kg mol$^{-1}$) to approximately

100 μL. A stock solution of CuSO$_4$(H$_2$O)$_5$ (12 mM) was mixed in a 1:1 ratio with a stock solution of THPTA (60 mM) and 20 μL were added to the concentrated dispersion. After 15 min the mixture was diluted to 2 mL, filtered by spin-filtration (Amicon Ultra-4, $M_w$ = 100 kg mol$^{-1}$) and suspended in freshly prepared 10 μM EDTA-free MOPS-buffer (Serva) solution (pH 7.4). The SWNT dispersion was adjusted to an optical density of 0.1 cm$^{-1}$ at the $E_{11}$ transition and aliquots of PP$_i$ in ultra-pure water were added to adjust the final concentration of pyrophosphate. A schematic of the workflow can be found in Supplementary Fig. 27. For the detection of PP$_i$ in spiked cell lysate and PP$_i$ released in a PCR the SWNT dispersion was adjusted to an optical density of approximately 0.02 cm$^{-1}$ at the $E_{11}$ transition in a final measurement volume of 100 μL. SWNT con-centrations down to ≈ 0.034 mg L$^{-1}$ (corresponding to an optical density of 0.02 cm$^{-1}$ at the $E_{11}$ transition for (6,5) SWNTs) were sufficient to achieve a high signal-to-noise ratio.

## Preparation of cell cultures

HEK293 cells were purchased from DSMZ German Collection of Microorganisms and Cell Cultures (ACC 305) and cultivated according to the supplier's protocol in a humidified 5% CO$_2$ atmosphere at 37 °C in T-75 flasks (Sarstedt) with a sub-cultivation ratio of 1:5 every 3-4 days. Cells were grown in 16 mL DMEM (Thermo Fisher Scientific) supple-mented with fetal bovine serum (FBS) (10%), penicillin (100 units mL$^{-1}$), and streptomycin (100 μg mL$^{-1}$, Thermo Fisher Scientific). Cells were seeded in 100 mm dishes (Sarstedt) at a density of 2.2 × 10$^6$ cells and grown for 4 days until confluency. Prior to cell harvest, the cultivation media was removed and cells were carefully washed twice with ultra-pure water. Cells were harvested by scraping in 200 μL ultra-pure water, immediately transferred to 1.5 mL Eppendorf cups and shock frozen in liquid nitrogen. Lysis was performed by pulsed sonication (1 s on / 1 s off) at 50% amplitude for 2 min (Qsonica Model Q700 with cuphorn attachment), followed by centrifugation at 21,000 × $g$ for 2 min to remove cell debris and large particles. The supernatant was used for further experiments.

## PCR reaction

All constructs created in this study were generated by standard PCR techniques. PCR reaction mix was assembled on ice in a total volume of 150 μL. The PCR reaction formulation was according to the Q5 High-Fidelity DNA polymerase manufacturer's (New England BioLabs, M0491S) instructions with 200 μM final concentration of dNTPs (10 mM dNTP mix, Promega, U1511) and 0.5 μM of each forward (T7pro-fwd TAATACGACTCACTATAGGGG) and reverse (T7term-rev TGCTAGTTATTGCTCAGCGG) primer (Eurofins Genomics). 150 ng of pET27b(+) vector with *E. coli ompc* gene served as template DNA. The mastermix was split into three individual sets and went through PCR cycling in an Eppendorf Mastercycler for 12 cycles, 24 cycles, or kept on ice, respectively. PCR program was a standard program with initial denaturation at 98 °C for 30 s, followed by 30 s annealing at 50.9 °C and elongation at 72 °C for 35 s for the aforementioned 12 or 24 cycles. PCR reactions were analyzed by gel electrophoresis (5 V cm$^{-1}$ for 45 min) on a 1% agarose TAE gel containing 1X GelRed (Biotium, 41003-T) and visualized under ultraviolet light by Bio-Rad ChemiDoc Gel Imaging System.

## Reporting summary

Further information on research design is available in the Nature Portfolio Reporting Summary linked to this article.

## Data availability

The datasets generated during and/or analyzed during the current study are available in the heiDATA repository (https://doi.org/10.11588/data/UOE7KX) and from the corresponding authors upon request. Source data are provided with this paper.

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

## Acknowledgements

This project has received funding from the European Research Council (ERC) under the European Union's Horizon 2020 research and innovation programme (Grant Agreement No. 817494 "TRIFECTs", S.S., J.Z.). C.A.S., S.J., and S.K. acknowledge funding by the Deutsche Forschungsgemeinschaft (DFG, German Research Foundation) under Germany's Excellence Strategy—EXC 2033—390677874 – RESOLV, the "Center for Solvation Science ZEMOS" funded by the German Federal Ministry of Education and Research BMBF and by the Ministry of Culture and Research of Nord Rhine-Westphalia, and the Volkswagen Stiftung. B.S.F. and H.L. gratefully acknowledge support by the DFG under grant numbers FL 834/5-1, FL 834/9-1, and FL 834/12-1.

## Author contributions

S.S. prepared and measured all samples and analyzed the data. C.A.S., S.J. and S.K. designed and conceived spiking and PCR experiments. E.M. performed synthesis and characterization of diazonium salts under supervision of A.S.K.H. H.L. and B.S.F. provided ATPE-sorted (6,5) SWNTs. J.Z. conceived and supervised the project. S.S. and J.Z. wrote the manuscript. All authors discussed the data analysis and commented on the manuscript.

## Funding

## Competing interests

The authors declare no competing interests.
