## [Peer Review File · Nature Communications]

Ratiometric fluorescent sensing of pyrophosphate with sp^3 -functionalized single-walled carbon nanotubesREVIEWER COMMENTS

Reviewer #1 (Remarks to the Author):

The paper presents a novel approach for the optical detection and quantification of inorganic pyrophosphate (PPi), a critical molecule in various biological processes, including DNA synthesis and cellular metabolism. The authors employ (6,5) single-walled carbon nanotubes (SWNTs) as near-infrared (NIR) luminescent probes. These SWNTs are functionalized with sp³ defects of covalently attached aryl alkyne groups. The scheme is based on the immobilization of Cu²⁺ ions on the SWNT surface promoted by the aryl alkyne groups, which leads to fluorescence quenching via photoinduced electron transfer. This quenching is reversed in the presence of copper-complexing analytes such as PPi, allowing for its detection.

The study is distinct for its ratiometric measurement approach, relying on the intensity ratio of defect-induced E11* emission to the E11 emission, which enables highly robust and quantitative detection of PPi across a broad concentration window. Further, the authors demonstrate the biocompatibility of their probe by coating the SWNTs with a phospholipid-polyethylene glycol layer, enabling the detection of PPi in complex biological environments like cell lysate. They also highlight the probe's potential utility in monitoring DNA synthesis in polymerase chain reactions (PCR).

Overall, the paper proposes a robust, ratiometric, and NIR luminescent probe for PPi detection, and outlines its applications in biological systems. This work could serve as a foundational platform for the design of future nanotube-based biosensors.

The paper is clearly written and well-presented. Still, prior to publication, the following points should be addressed:

1. The working concentrations of pyrophosphate reported by the authors are in hundreds of micromolar to millimolar. Does this limit practical applications? What is the limit of detection?
2. The authors have introduced a 1:1 mixture of copper(II) sulfate pentahydrate (CuSO₄·5H₂O) and tris(3-hydroxypropyl-triazolylmethyl)amine (THPTA) to quench the photoluminescence of the defect-induced SWCNTs. However, the manuscript lacks thorough characterization of the complex formed between CuSO₄·5H₂O and THPTA. Based on the information provided in the manuscript, it appears that Cu²⁺ ions (sourced from copper sulfate) and THPTA form a mixture rather than a well-defined complex. Characterization of this complex would offer a comprehensive understanding of the metal ion replacement reaction, which constitutes the basis of the current study. Also, determining the oxidation state of copper in this complex is important to substantiate the quenching mechanism, which was ascribed to photoinduced electron transfer.
3. The addition of Cu(THPTA), which the authors have denoted as "[Cu]," to SWCNTs results in the quenching of fluorescence. Moreover, there is an observed peak shift in the E11 and E11* peaks of SWCNTs upon the addition of [Cu]. The authors should further discuss possible factors that potentially contribute to such a shift.
4. The fluorescence quenching of SWCNTs, upon addition of [Cu], is attributed to photoinduced electron transfer, which is logical given that the copper can accept electrons from donor ligands. It is surprising to observe, as indicated in Figure S11, that the extent of quenching of SWCNTs (specifically the E11* peak) is relatively similar when using copper, nickel, and cobalt ions as quenchers, given that they have different electron occupancy, which should influence their capacity to accept electrons. Do the authors have a hypothesis regarding this point?
5. The authors have used luminescence lifetime analysis to dismiss the possibility of ground-state quenching. Still, is it possible that both ground state and excited state phenomena contribute to the observed quenching?
6. The design strategy of the detection technique relies on the chemical interaction between copper ions and pyrophosphate. There is a finite possibility of copper pyrophosphate formation in this process, which is water-insoluble. Did the authors find insoluble copper pyrophosphate? It would provide valuable insights into the detection mechanism.
7. The authors have made an attempt to detect pyrophosphate in a mixture of ATP and pyrophosphate. They assert that they have successfully detected pyrophosphate in this complex mixture. However, Figure 3 reveals that both ATP and pyrophosphate restore the fluorescence of [Cu]-

quenched SWCNTs. Could the detection be an additive effect arising from both ATP and pyrophosphate?

8. As per the claims on the restoration of fluorescence by PPI, copper interacts with pyrophosphate in preference to the chemical interactions that keep them bound to the surface of the SWCNTs. Can the authors compare the association constant between copper ions and THPTA vis-à-vis copper ions and PPI?

9. Did the author test the effect of PPI on the fluorescence of SWCNTs, i.e., without the addition of [Cu]? It would be an important control experiment.

10. The authors have claimed successful incorporation of covalent defects using fluorescence measurement. It would be intriguing to explore alternative techniques for confirming this phenomenon, like Raman. I leave this point as a choice of the authors.

Reviewer #2 (Remarks to the Author):

Settele et al developed a ratiometric fluorescent nanosensor that can quantify inorganic pyrophosphate (PPI) in vitro. Non-covalently functionalized a $\text{CuSO}_4(\text{H}_2\text{O})_5/\text{THPTA}$ complex quenches the PL of aryl alkyne functionalized (6,5)-single-walled carbon nanotubes (SWCNTs) via photoinduced energy transfer. The SWCNT PL recovers in the presence of copper-complexing analytes such as pyrophosphate. Photophysical studies were conducted to understand the PPI-induced quenching (sensing mechanism). To demonstrate potential applications, the nanosensors were coated with phospholipid-PEG and tested the functionality for the detection of PPI in cell lysate and for monitoring the progress of DNA synthesis in a polymerase chain reaction. The manuscript presents interesting results of rational design of new CNT-based biosensors and could become publishable after the following criticisms have been addressed.

1. The authors highlighted the potential in vivo and live cell applications of the sensor technology. Reversible detection is key to dynamically monitor biological processes in such systems. In the current sensor design, once PPI complexes with all the physisorbed [Cu] on SWCNT sidewall, the SWCNT PL will no longer respond to PPI and the response may not be linear. The authors should clarify in the discussion section, the limitations and potential improvement of the sensor technology towards applications in live cell and in vivo studies.

2. SWCNT PL responds to local dielectric environments and the E11 and E11* exhibit different sensitivity. This may impact the ratiometric intensity-based quantification of PPI. The authors should comment on this possibility and the need for obtaining calibration curves for different systems in the discussion section.

3. The reviewer is wondering if the concentration of SWCNTs would change the sensitivity to PPI. If this is the case, the authors should note this parameter and provide the optimal concentration of SWCNTs for PPI sensing.

4. As the weak coordination between the aryl alkyne defects and [Cu] is associated with more sensitive quenching behavior of E11* PL, it is possible that defect density on the nanosensor sensitivity to PPI and dynamic range. The authors should include the discussion and/or additional experiments to provide better guidance on the sensor design.

5. Discussion of pros and cons compared to other PPI sensors should be included.

6. Photophysical mechanistic studies on signal transduction are interesting. The authors should comment on how the wavelength shifts of E11 and E11* PL were observed upon [Cu] adsorption.

7. The authors should note the difference in the spectral shape between Figs 5e,f and others. Using a different spectrometer won't impact the spectral shapes if both are correctly calibrated. The concentration may not significantly impact the peak shapes as well if the PL spectra were measured in the dilute regime (Powell, J Phys Chem Lett 2018)

Reviewer #3 (Remarks to the Author):

In this manuscript, the authors describe a biosensor for pyrophosphate (PP) based on sp³-functionalized single-walled carbon nanotubes (SWNTs). The SWCNTs were engineered to reversibly complex copper ions, resulting in a quenching of fluorescence that would be recovered in the presence of PP. The authors further explore the underlying mechanism and application of these sensors to cell lysate and PCR.

There are only a few examples that use sp³-defect fluorescence for optical SWNT sensing. These sensors hold great promise for improving SWNT sensor response and designing next-generation optical sensors more rationally. The manuscript is very well written, with all important controls considered. It lacks only a few details on the description and analysis. I therefore recommend publishing with minor revisions.

Request for additional information:

1. In the introduction, the authors focus on the advantage of better PLQY for sp³-defected SWNTs. It would further motivate this study to mention potential applications for multimodal sensing. The sp³ defect not only brightens SWNTs but also forms new peaks at different wavelengths that could be useful for multi-analyte detection.
2. The authors should specify the physiologically relevant concentration of PP in biological systems.
3. In the experiment of copper-induced fluorescent quenching, both E11 and E11* peaks show a decrease in PL. The authors should compare the extent of E11 quenching to that of unmodified (6,5) SWNTs. This comparison could be useful in understanding the underlying mechanism as to whether the quenching of the E11 peak is entirely due to Cu or if it is affected by the E11* peak.
4. The authors propose a hypothesis for the quenching, but little discussion on the shifting. It would strengthen the manuscript to propose a hypothesis (or several hypotheses) for the shifting response (for example, possible conformation effects?), even if this effect is more elusive to study.

Recommendations on tables and figures:

5. Fig 4f is a bit confusing. I suggest redrawing this figure with several equations (such as to indicate binding and release) in the discussion to explain this process more clearly.
6. SI Fig 1 is quite important, and I would recommend to move it to the main manuscript.
7. Fig. 2b caption should clarify that the plot is normalized to the E11 peak intensity.
8. In Fig 2f, I suggest directly using peak shift rather than peak position.
9. In Fig 5e and 5f, the wavelength of the axis should be the same as 5a and 5b.
10. Table S1 should list all the structures of the analytes to help identify possible relationships between some functional groups and responses.
11. Corrections recommended for a few minor typos: "into an aqueous dispersions" (singular vs. plural) and spacing needed for ">23uM"

Reviewer #4 (Remarks to the Author):

Reviewer #5 (Remarks to the Author):

Article Review: Ratiometric fluorescent sensing of pyrophosphate with sp³-functionalized single-walled carbon nanotubes

Noteworthy Results:

- The immobilization and displacement of Cu²⁺ ions on sp³ functionalized (6,5) SWNTs serve as effective near-infrared luminescence probes. Fluorescent quenching occurs when Cu²⁺ ions are immobilized near the SWNT and is restored when the ions are displaced, facilitating the detection of Cu²⁺ complexing analytes. This sensor exhibits some preference for pyrophosphate (PPi), a molecule integral to various biological processes including DNA synthesis and cellular metabolism.
- Diazonium salt covalent functionalization introduces a quantum defect, enabling a ratiometric sensor reading. The sensor exhibits spectral modulation (wavelength shift and PL quenching/enhancement) upon exposure to the target analyte, with a more pronounced response in the E11* peak than the E11 peak. The presence of the E11 peak serves as an embedded reference within the sensor.
- The sensor retains its functionality when transitioning the non-covalently bound corona from SDS to PL-PEG5000. The latter's biocompatibility enables the detection of analytes in biological environments. Consequently, this adaptation of the sensor facilitates the monitoring of PPi in cell lysate during a polymerase chain reaction.

Originality: High

- The sensing scheme is novel, integrating both organic color centers (quantum defects) and Cu²⁺ ion immobilization to achieve sensing, a combination not previously reported to my knowledge.

Broad Appeal: Moderate

- This study could motivate further research exploring innovative ligand systems and metal-ion immobilization techniques for diverse applications.

Clarity: Top 10%

- Incorporating an explanation of amplitude average lifetime prior to introducing its results would enhance those results' comprehensiveness.

Revisions:

1) Page 8, lines 172-173: The authors attribute the observed drop in photoluminescence, resulting in the final data point's deviation from the trend, to particle aggregation at elevated analyte concentrations. To substantiate this assumption, it would be good to measure the particle size distribution correlating with varying analyte concentrations.

2) Page 17, lines 354-358: The authors postulate that the reduction in PL quenching, observed when comparing SDS to PL-PEG5000, could be attributed to variations in SWNT surface coverage. To strengthen this assertion, the authors should either cite literature that supports this claim or employ an experimental technique to actually measure such surface coverage. A technique like Molecular Probe Absorption has been previously used to quantify the surface coverage of the non-covalently absorbed corona on the carbon nanotube surface (<https://doi.org/10.1021/acs.nanolett.9b02647>). The referenced technique has previously been utilized to actually measure the surface coverage of SDS functionalized SWNTs, the results placing it in the middle of surface coverage when compared to the other reported SWNT coronas; as such it is hard to tell without experiments or literature whether PL-PEG5000 will have more SWNT surface coverage.

Point-by-Point Response

Manuscript # NCOMMS-23-40277

We thank all reviewers for their careful consideration of our manuscript. We have addressed all of their questions and points they raised with additional data, figures, references and discussion as outlined in detail in the following point-by-point response. Changes we applied to the revised manuscript or the supplementary information are highlighted in **bold**. The revised versions of the manuscript and supplementary information with marked changes are provided separately for review.

REVIEWER COMMENTS

Reviewer #1 (Remarks to the Author):

The paper presents a novel approach for the optical detection and quantification of inorganic pyrophosphate (PPi), a critical molecule in various biological processes, including DNA synthesis and cellular metabolism. The authors employ (6,5) single-walled carbon nanotubes (SWNTs) as near-infrared (NIR) luminescent probes. These SWNTs are functionalized with sp³ defects of covalently attached aryl alkyne groups. The scheme is based on the immobilization of Cu²⁺ ions on the SWNT surface promoted by the aryl alkyne groups, which leads to fluorescence quenching via photoinduced electron transfer. This quenching is reversed in the presence of copper-complexing analytes such as PPi, allowing for its detection. The study is distinct for its ratiometric measurement approach, relying on the intensity ratio of defect-induced E11* emission to the E11 emission, which enables highly robust and quantitative detection of PPi across a broad concentration window. Further, the authors demonstrate the biocompatibility of their probe by coating the SWNTs with a phospholipid-polyethylene glycol layer, enabling the detection of PPi in complex biological environments like cell lysate. They also highlight the probe's potential utility in monitoring DNA synthesis in polymerase chain reactions (PCR). Overall, the paper proposes a robust, ratiometric, and NIR luminescent probe for PPi detection, and outlines its applications in biological systems. This work could serve as a foundational platform for the design of future nanotube-based biosensors. The paper is clearly written and well-presented. Still, prior to publication, the following points should be addressed:

1. The working concentrations of pyrophosphate reported by the authors are in hundreds of micromolar to millimolar. Does this limit practical applications? What is the limit of detection?

RESPONSE:

We calculated the limit of detection (LOD) for PP_i based on the standard deviation (SD) of the response upon addition of [Cu] to the SWNT sensor (SD_{blank} ; $n = 10$) and the slope m_{linear} from a linear fit of the calibration curve at low PP_i concentrations. SD_{blank} was measured for the same sample as shown in Figure 5.

$$LOD = (3 * SD_{blank})/m_{linear}$$

This analysis yielded a LOD for the different sensing parameters of:

- E_{11} intensity: LOD = 5.2 μM
- E_{11}^* intensity: LOD = 12.5 μM
- E_{11}^*/E_{11} PL ratio: LOD = 4.4 μM

We included this information in the revised manuscript as new Supplementary Fig. 23.

New Supplementary Fig. 23. **a** PP_i concentration dependent changes of the E_{11}^*/E_{11} PL intensity ratio with linear fit to data at low concentrations of PP_i ($m_{linear} = 0.00381 \mu\text{M}^{-1}$). **b** Normalized PL spectra of 4-ethynylbenzene-functionalized (6,5) SWNTs dispersed in PL-PEG₅₀₀₀ and 10 mM MOPS buffer after addition of [Cu] with a standard deviation of the E_{11}^*/E_{11} PL intensity ratio of $SD_{blank} = 0.0056$ ($n = 10$). The displayed data correspond to the data set shown in Fig. 5.

With an LOD of 4.4 μM for the PL intensity ratio, many biologically and clinically relevant conditions can be diagnosed. For example, patients with hypophosphatasia show a significantly increased level of PP_i with concentrations between 5 and 17.5 μM in plasma and 45-234 μM in urine (*J. Clin. Invest.* **1971**, *50*, 961-969; *Lancet* **1965**, *286*, 461-464; *PLoS ONE* **2017**, *12*, e0180098; *Clin. Chim. Acta.* **2001**, *314*, 187-194). Furthermore, the intracellular PP_i

concentration plays a key role in the mitochondrial metabolism with concentrations of 70-120 μM reported for rat liver cells (*Am. J. Physiol. Cell Physiol.* **2001**, *281*, C1–C11; *Arthritis & Rheumatism* **2000**, *43*, 1560-1570). Various rheumatological conditions also show increased PP_i concentration in synovial fluids with concentrations ranging from 3 to 34 μM . (*Arthritis & Rheumatism* **1988**, *31*, 408-413)

While the LOD reported here is close to most PP_i concentrations relevant for many clinical conditions we would like to highlight that the presented sensor is not limited to the detection of PP_i but can also detect other phosphates such as ATP. As PP_i is usually accompanied by other phosphates the total concentration that leads to a response of the SWNT sensor is expected to be higher and thus should be used mainly to detect dynamic changes in the relevant concentration levels of PP_i . The ability to track, *e.g.*, the hydrolysis reaction of ATP to PP_i during PCR was demonstrated in Fig. 6.

In summary, the LOD and even more importantly the dynamic range of the sensor is in the relevant range for most biologically relevant scenarios (1 μM - 100 μM).

We extended the discussion of this topic in the revised manuscript on page 19.

2. The authors have introduced a 1:1 mixture of copper(II) sulfate pentahydrate ($\text{CuSO}_4 \cdot 5\text{H}_2\text{O}$) and tris(3-hydroxypropyl-triazolylmethyl)amine (THPTA) to quench the photoluminescence of the defect-induced SWCNTs. However, the manuscript lacks thorough characterization of the complex formed between $\text{CuSO}_4 \cdot 5\text{H}_2\text{O}$ and THPTA. Based on the information provided in the manuscript, it appears that Cu^{2+} ions (sourced from copper sulfate) and THPTA form a mixture rather than a well-defined complex. Characterization of this complex would offer a comprehensive understanding of the metal ion replacement reaction, which constitutes the basis of the current study. Also, determining the oxidation state of copper in this complex is important to substantiate the quenching mechanism, which was ascribed to photoinduced electron transfer.

RESPONSE:

The complex formed by CuSO_4 and THPTA is an already well-studied Cu(II) complex that is frequently used in organic chemistry. Its oxidation state has been confirmed previously by cyclic voltammetry measurements (*Org. Lett.* **2004**, *6*, 2853–2855) to be Cu(II) and a multitude of similar ligands have been described with similar properties. (see *e.g.* *J. Am. Chem. Soc.* **2011**,

133, 17993–18001; *Inorg. Chem.* **2003**, *42*, 5267–5273) The crystal structure of other very similar tripodal ligands were previously reported (see *Acta Cryst.* **1997**, *C53*, 559-562).

A close analogue Cu(II)-TBTA (soluble in organic solvents) is commercially available (*e.g.* by *Lumiprobe*). It forms a well-defined complex with Cu in the +2 oxidation state. Due to the widespread use of this complex in literature, we believe that further characterization is not necessary.

We added the available information about the complex in the revised manuscript on page 6. We also provide additional absorption spectra of the formed complex (see comment 8.)

3. The addition of Cu(THPTA), which the authors have denoted as "[Cu]," to SWCNTs results in the quenching of fluorescence. Moreover, there is an observed peak shift in the E₁₁ and E₁₁* peaks of SWCNTs upon the addition of [Cu]. The authors should further discuss possible factors that potentially contribute to such a shift.

RESPONSE:

As mentioned by **reviewer 3** (see below) this effect (peak shifts) is more elusive to study than PL quenching. Based on our measurements shown in Fig. 4a, the wavelength shifts of the E₁₁ and E₁₁* peaks only occur when CuSO₄ is complexed by THPTA and thus adsorbed to the SWNT sidewall. We hypothesize that the increased concentration of Cu²⁺ ions in close proximity to the pristine SWNT lattice and the defect site leads to a bathochromic shift induced by solvatochromism, which can be understood as an electrostatic stabilization of the photoexcited electron-hole pair by molecules in close proximity to the SWNT (*e.g.* the solvent). Shiraki *et al.* showed that such shifts also occur for defect state emission (*Chem. Commun.* **2019**, *55*, 3662). Larsen *et al.* have reported that solvatochromic shifts depend on the solvent polarity and the solvation of the SWCNT (*J. Am. Chem. Soc.* **2012**, *134*, 12485). Solvent polarity is expected to increase upon adsorption of Cu²⁺ ions and solvation may change due to the additional hydration shell surrounding the Cu²⁺ ions. Furthermore, coordination of Cu²⁺ ions to sulfate (in case of SDS) or phosphate groups (in case of PL-PEG₅₀₀₀) can alter the structure of the surfactant surface coverage. However, we expected this effect to be minor due to the absence of wavelength shifts when Cu²⁺ ions are not complexed by THPTA.

In summary, our explanation for the observed shifts are (local) changes in solvation that affect the exciton energy as well as the excitation decay pathways.

We added a short discussion about the possible origins of the observed wavelength shifts in the revised manuscript on page 12.

4. The fluorescence quenching of SWCNTs, upon addition of [Cu], is attributed to photoinduced electron transfer, which is logical given that the copper can accept electrons from donor ligands. It is surprising to observe, as indicated in Figure S11, that the extent of quenching of SWCNTs (specifically the E11* peak) is relatively similar when using copper, nickel, and cobalt ions as quenchers, given that they have different electron occupancy, which should influence their capacity to accept electrons. Do the authors have a hypothesis regarding this point?

RESPONSE:

First, we would like to point out that the comparison of the different metal ions was performed without THPTA ligand since the complexation of THPTA to Ni(II) and Co(II) ions may vary significantly. The effect of metal ions on the PL properties of SWNT is rather elusive and has been reported several times without any proposals for a detailed quenching mechanism. As described in the manuscript we suspect photo-induced electron transfer (PET) to be the most dominant quenching mechanism. Consequently, a close correlation with the standard reduction potentials of the metal ions is expected. These are very similar for Ni(II/0) and Co(II/0) with -0.26 V and -0.28 V respectively (*J. Phys. Chem. Ref. Data.* **1989**, *18*, 1-21). In both cases, a negative ΔG_{PET} of -1.61 and -1.65 eV can be calculated and makes the PET process very likely. In contrast to that, Cu(II/0) and Cu(II/I) have a standard reduction potential of 0.339 and 0.161 V respectively (ΔG_{PET} of -2.23 and -2.05 eV) (*J. Phys. Chem. Ref. Data.* **1989**, *18*, 1-21). Thus, similar quenching behaviour for Ni(II) and Co(II) ions can be expected based on their reduction potentials while Cu(II) ions may lead to stronger PL quenching. However, further studies with more metal ions are necessary to give a more conclusive picture.

We extended our discussion about the quenching mechanism in the revised Supplementary Information in the section “Estimation of PET & mechanistic considerations”.

5. The authors have used luminescence lifetime analysis to dismiss the possibility of ground-state quenching. Still, is it possible that both ground state and excited state phenomena contribute to the observed quenching?

RESPONSE:

We can rule-out the possibility of ground-state effects due to the close correlation of PL lifetime and PL quenching measurements. Based on the recorded PL spectra a reduction in PL lifetime of ~ 1.91 is expected and corresponds very well (within the experimental limits) to the measured reduction of ~ 2.25 (see **Table 1**). Ground-state effects would not contribute to changes of the PL lifetimes and thus a stronger discrepancy between the two values would be expected (see **new Supplementary Fig. 13c**). Furthermore, we do not observe any bleaching effects in the measured absorption spectra upon the addition of [Cu], which would be a typical indicator for ground-state effects (see Supplementary Fig. 12). The only exception to this rule is when a heterogeneous fluorophore system is present where emission originates, *e.g.*, from two different fluorophores and ground state quenching is selective for one species (see *Photochem. Photobiol.* **1998**, 67, 475). While each sp^3 defects is likely to possess an individual lifetime, the differences should be small and defect state dynamics are expected to be similar. Hence, sp^3 defects can be treated as a homogeneous fluorophore system and we do not consider ground-state bleaching as relevant in our system.

6. The design strategy of the detection technique relies on the chemical interaction between copper ions and pyrophosphate. There is a finite possibility of copper pyrophosphate formation in this process, which is water-insoluble. Did the authors find insoluble copper pyrophosphate? It would provide valuable insights into the detection mechanism.

RESPONSE:

The formation of the $Cu(II)_2P_2O_7$ complex (which is insoluble in water) is usually only observed at very low pH values. For near neutral pH values (as in our experiments that were conducted in a buffer system) the dominant species is $Cu(II)(P_2O_7)_2$ hydrate, which is highly soluble in water (*Inorg. Chim. Acta* **1968**, 2, 74-80; *J. Phys. Chem. A* **1997**, 101, 5131–513). As a consequence, no insoluble $Cu(II)_2P_2O_7$ can be expected when the sensor is used at neutral pH

in a buffer system. During our studies conducted in SDS, where $\text{Cu(II)}_2\text{P}_2\text{O}_7$ formation might be possible (pH \sim 4.8), no visible precipitation was observed.

7. The authors have made an attempt to detect pyrophosphate in a mixture of ATP and pyrophosphate. They assert that they have successfully detected pyrophosphate in this complex mixture. However, Figure 3 reveals that both ATP and pyrophosphate restore the fluorescence of [Cu]-quenched SWCNTs. Could the detection be an additive effect arising from both ATP and pyrophosphate?

RESPONSE:

We agree with the reviewer. The response observed upon addition of a mixture of ATP and PP_i is an additive effect even though there is a selectivity for PP_i (Figure 3a). This effect is then used to track the progress in PCR as shown in Fig. 6.

We have clarified our statement in the revised manuscript on page 11.

8. As per the claims on the restoration of fluorescence by PP_i , copper interacts with pyrophosphate in preference to the chemical interactions that keep them bound to the surface of the SWCNTs. Can the authors compare the association constant between copper ions and THPTA vis-à-vis copper ions and PP_i ?

RESPONSE:

The easiest way to compare the association constant (or dissociation constant) between Cu^{2+} ions and THPTA and PP_i is titration of a Cu/THPTA complex with PP_i and track *in-situ* changes of relevant absorption bands. For this purpose, the absorption band around \sim 315 nm is suitable as it arises from the THPTA ligand only when coordinated to CuSO_4 . Neither the THPTA ligand itself nor CuSO_4 show significant absorption bands at this wavelength. Upon titration with PP_i , this absorption feature vanishes with increasing PP_i concentration, indicating the removal of Cu^{2+} ions from the THPTA ligand. No absorption feature for the $\text{PP}_i/\text{Cu}^{2+}$ complex is expected at this absorption wavelength due to the lack of an aromatic backbone. Hence, the concentration of the Cu/THPTA complex can be directly correlated to the absorbance at 315 nm. When fitted with a Hill-function (see New Supplementary Fig. 8 below), a dissociation constant of $K_d = 20.3 \times 10^{-6}$ M can be extracted. The corresponding K_d extracted from PL measurements is 25

times higher ($K_d = 516 \times 10^{-6} \text{ M}$) than that of the free complex, indicating that $\text{Cu}^{2+}/\text{PP}_i$ complex formation is weakened due to steric shielding of the Cu^{2+} centre by adsorption to the SWNT surface.

We included this information in the revised Supplementary Information as new Fig. 8.

New Supplementary Fig. 8. **a** Absorption spectra of a 1:1 complex of CuSO_4 and THPTA (12.5 mM), CuSO_4 (12.5 mM) and THPTA (12.5 mM) **b** Absorption spectra of a 1:1 complex of CuSO_4 and THPTA (15 μM) after addition of various concentrations of PP_i . **c** Black squares: PP_i concentration dependent changes of the absorption at 315 nm are displayed with A as the absorbance and A_0 the absorbance before the addition of PP_i . Red circles: PP_i concentration dependent changes of the E_{11}^*/E_{11} PL intensity ratio extracted from PL measurements as displayed in Fig. 2. R is the E_{11}^*/E_{11} intensity ratio and R_0 the E_{11}^*/E_{11} intensity ratio after addition of $[\text{Cu}]$. The data were fitted with a Hill-function and dissociation constants of $20.3 \cdot 10^{-6} \text{ M}$ and $516 \cdot 10^{-6} \text{ M}$ were extracted, respectively.

9. Did the author test the effect of PP_i on the fluorescence of SWCNTs, i.e., without the addition of [Cu]? It would be an important control experiment.

RESPONSE:

Yes, we performed this control experiment and conducted PL measurements of 4-ethynylbenzene functionalized SWCNTs dispersed in SDS and PL-PEG₅₀₀₀ after the addition of various concentrations of PP_i . We did not observe an increase in PL intensity or PL ratio in any of these measurements. For SDS-dispersed SWCNTs a small decrease in overall PL intensity as well as E_{11}^*/E_{11} ratio can be observed for low concentrations of PP_i (<2 mM) and a significant drop at high concentrations of PP_i (>11 mM). This decrease in PL intensity is accompanied by a small red shift of E_{11} and E_{11}^* emission. At such high salt concentrations aggregation effects are highly likely for surfactant dispersed SWCNTs and can explain both observed effects (see *J. Am. Chem. Soc.* **2007**, *129*, 1898–1899; *Langmuir* **2014**, *30*, 10899–10909). A similar effect can be observed in Fig. 2a and 2b, although to a lesser extent. As we typically stayed below a PP_i concentration of <4 mM and the E_{11}^*/E_{11} PL ratio is affected only slightly it should not alter the evaluation of the presented data.

We agree with the reviewer that this is a very important control experiment and have included this data set in the revised Supplementary Information as new Fig. 3 and Fig. 18.

New Supplementary Fig. 3. Absolute (a) and normalized (b) PL spectra of 4-ethynylbenzene-functionalized (6,5) SWCNTs dispersed in SDS (without [Cu]) before and after the addition of various concentrations of PP_i .

New Supplementary Fig. 18. Absolute (a) and normalized (b) PL spectra of 4-ethynylbenzene-functionalized (6,5) SWNTs dispersed in PL-PEG₅₀₀₀ before and after the addition of various concentrations of PP_i.

10. The authors have claimed successful incorporation of covalent defects using fluorescence measurement. It would be intriguing to explore alternative techniques for confirming this phenomenon, like Raman. I leave this point as a choice of the authors.

RESPONSE:

It is well known that the introduction of luminescent defects in functionalized SWNTs *via* diazonium salts leads to an increase of the disorder-induced Raman D-mode. (*Nat. Chem.* 2013, 5, 840–845, *J. Phys. Chem. Lett.* 2022, 13, 3542–3548). **Nevertheless, within the scope of comment 4 by reviewer 2 we now provide Raman spectra of pristine and functionalized (6,5) SWNTs in the revised Supplementary Information as Supplementary Fig. 15g.**

New Supplementary Figure 15g. Normalized averaged Raman spectra of (6,5) SWNTs functionalized with 4-ethynyl benzene diazonium tetrafluoroborate at various concentrations leading to an increase in Raman D-Mode (see zoom-in).

Reviewer #2 (Remarks to the Author):

Settele et al developed a ratiometric fluorescent nanosensor that can quantify inorganic pyrophosphate (PPi) in vitro. Non-covalently functionalized a $\text{CuSO}_4(\text{H}_2\text{O})_5/\text{THPTA}$ complex quenches the PL of aryl alkyne functionalized (6,5)-single-walled carbon nanotubes (SWCNTs) via photoinduced energy transfer. The SWCNT PL recovers in the presence of copper-complexing analytes such as pyrophosphate. Photophysical studies were conducted to understand the PPi-induced quenching (sensing mechanism). To demonstrate potential applications, the nanosensors were coated with phospholipid-PEG and tested the functionality for the detection of PPi in cell lysate and for monitoring the progress of DNA synthesis in a polymerase chain reaction. The manuscript presents interesting results of rational design of new CNT-based biosensors and could become publishable after the following criticisms have been addressed.

1. The authors highlighted the potential in vivo and live cell applications of the sensor technology. Reversible detection is key to dynamically monitor biological processes in such systems. In the current sensor design, once PPi complexes with all the physisorbed [Cu] on SWCNT sidewall, the SWCNT PL will no longer respond to PPi and the response may not be linear. The authors should clarify in the discussion section, the limitations and potential improvement of the sensor technology towards applications in live cell and in vivo studies.

RESPONSE:

We agree with the reviewer and addressed the limitations and potential improvements of the sensor in the discussion section of the revised manuscript (page 24).

For many biological questions, reversible sensing is necessary if the goal is to follow dynamic changes over time. The current sensor design causes an irreversible change in fluorescence in the presence of PPi. This is a universal feature of metal-displacement assays. In this detection scheme reversibility might only be achieved by addition of new Cu^{2+} ions or design of a new immobilization strategy where Cu^{2+} complexation by PPi only impacts its distance to the SWNT surface and thus degree of PL quenching. However, irreversible sensing modalities also have advantages. For example, assays to identify the presence of a pathogen (such as the COVID-19 lateral flow assays) work better if the interaction responsible for sensing is irreversible.

Additionally, fast dynamic changes and complex kinetics can make reversible sensor responses ambiguous. Nevertheless, it primarily depends on the envisioned application. In our work we wanted to measure in a ratiometric way the concentration of PP_i after a given time (see Fig. 6). In such end-point assays an irreversible sensor is more reliable because a reversible sensor would be biased by time-dependent PP_i hydrolysis.

Further, as outlined in the main text on page 13 and page 24, we expect great potential for this sensor to be used with other ligands that show similar properties as THPTA but display a higher sensitivity and/or selectivity towards PP_i or, e.g., ATP. We are currently exploring the possibility to use other ligands, however, can only provide this data for the review process as the properties of these ligands are still under investigation and the results are preliminary. For example, other triazole ligands such as BTTA could be used in the future. They show similarly strong SWNT PL quenching when complexed by Cu^{2+} ions (see below, left) but potentially exhibit a higher sensitivity/selectivity towards PP_i . Furthermore, not only ligands based on a triazole core structure could be used. Zhang *et al.* reported a phenanthroline ligand that shows very high selectivity towards PP_i even in the presence of other phosphates (*Chemistry Select* **2018**, 3, 10057). This system is also capable of immobilizing Cu^{2+} ions on the SWNT surface as we also observed a strong SWNT PL quenching (see below, right).

PL spectra of (6,5) SWNTs functionalized with 4-ethynylbenzene function groups after addition of Cu^{2+} /BTTA (left) and Cu^{2+} /phenanthroline complexes that may serve as starting points for further development of this SWNT sensor.

2. SWCNT PL responds to local dielectric environments and the E11 and E11* exhibit different sensitivity. This may impact the ratiometric intensity-based quantification of PP_i . The authors should comment on this possibility and the need for obtaining calibration curves for different systems in the discussion section.

RESPONSE:

The impact of different local environments on the ratiometric intensity-based quantification has been very low in our experiments. The detection scheme remained largely unchanged even after drastic changes of the environment, such as the transfer from SDS/H₂O to PL-PEG₅₀₀₀/buffer. However, it is clear that for each system a new calibration curve is necessary to correctly quantify PP_i and interpret, for example, shifts in PL peak position. The necessity to generate individual calibration curves before application is common for many well-established analyte detection schemes such as enzyme-linked immunosorbent assays (ELISA).

We addressed this issue in the revised manuscript on page 20.

3. The reviewer is wondering if the concentration of SWCNTs would change the sensitivity to PP_i. If this is the case, the authors should note this parameter and provide the optimal concentration of SWCNTs for PP_i sensing.

RESPONSE:

The SWNT concentration certainly has an impact on the sensor sensitivity. Such a change in sensitivity can be observed in Fig. 5. In the measurement displayed in Fig. 5f, approximately 1/5 of the typical SWNT concentration was used and thus upon addition of 188 μM PP_i we already observed a full recovery of the original PL signal. At this concentration of PP_i the SWNT PL is usually not completely restored as can be seen in Fig. 5a. These differences indicate that the dynamic range shifts towards lower concentrations of PP_i for lower concentrations of SWNTs. To achieve the highest sensitivity, it is advisable to lower the SWNT concentration as much as possible, while the PL signal remains at an acceptable signal to noise ratio. The lowest possible SWNT concentration that can be used is expected to depend on the experimental setup and PL quantum yield of the SWNT sensor. Furthermore, contingent on the detection range in which PP_i should be detected, the use of higher concentrations of SWNTs might be beneficial. In our experiments, SWNT concentrations of 0.04 - 0.17 mg L⁻¹ (with respect to (6,5) SWNTs) were sufficient to achieve a high signal to noise ratio.

We added this information on page 19, Figure 5 caption and in the Methods Section on page 28.

4. As the weak coordination between the aryl alkyne defects and [Cu] is associated with more sensitive quenching behavior of E_{11}^* PL, it is possible that defect density on the nanosensor sensitivity to PPI and dynamic range. The authors should include the discussion and/or additional experiments to provide better guidance on the sensor design.

RESPONSE:

We agree with the reviewer and have performed the corresponding experiments. The obtained results are presented in the revised Supplementary Information Fig. 15 (see below) and further discussion has been added.

New Supplementary Fig. 15. Normalized (a-c) and absolute (d-f) PL spectra of (6,5) SWNTs functionalized at various concentrations of 4-ethynyl benzene diazonium tetrafluoroborate in SDS after addition of 15 μM [Cu] and 4 mM PP_i . g Normalized Raman spectra of functionalized (6,5) SWNTs. h Quenching factor (QF) of the E_{11}^*/E_{11} intensity ratio vs. change in D/G⁺ area ratio ($\Delta D/G^+$). i Relative change in PL intensity ($I_{\text{PP}_i}/I_{\text{Cu}}$) of E_{11} and E_{11}^* emission vs. change in D/G⁺ area ratio ($\Delta D/G^+$).

PL spectra were recorded for three different defect densities on SWNTs in SDS dispersion. Absolute and normalized PL spectra are shown below. The defect density can be correlated with the differential Raman $\Delta D/G^+$ area ratio, which represents the change of the D/G^+ area ratio in comparison to the pristine sample. Upon addition of 15 μM $[\text{Cu}]$ similarly strong quenching and PL recovery after addition of 4 mM PP_i was observed, although with subtle differences. Extracting the quenching factor (QF, $I_{[\text{Cu}]} / I_{\text{Ref-Alkyne}}$) of the PL E_{11}^*/E_{11} intensity ratio and plotting it against the $\Delta D/G^+$ area ratio shows that the overall reduction of the PL ratio is slightly reduced at higher defect densities. This is reasonable, as more defect sites must be coordinated by Cu^{2+} ions in contrast to samples with low defect densities to achieve a similarly strong quenching of the defect state emission. However, we do not observe a significant change in the sensitivity/dynamic range as shown in Fig. S22a.

In contrast to this, a dependence of sensitivity on the absolute PL intensities was observed. When extracting the relative increase in PL intensities for E_{11} and E_{11}^* emission, *i.e.* the ratio of PL intensity before and after the addition of PP_i ($I_{\text{PP}_i} / I_{[\text{Cu}]}$), a clear connection to the defect density appears. A higher sensitivity was found for low defect densities. A similar effect is shown in Fig. S22b, where the dynamic range shifted towards higher concentrations of PP_i when SWNTs were functionalized to a higher degree.

In summary, the best sensitivity can be obtained at very low defect densities, however, its impact on the PL intensity ratio is minimal. Thus, we advise to use defect densities that are close or slightly below the PLQY maximum for functionalized SWNTs to obtain an overall high PL signal with good sensitivity.

5. Discussion of pros and cons compared to other PP_i sensors should be included.

RESPONSE:

We agree with the reviewer and added a brief comparison on page 20/21.

Here we consider only the comparison to other fluorescent PP_i sensors emitting in the NIR-II as relevant. So far only one report is available that could be used for such a direct comparison. Su *et al.* (*J. Mater. Chem. B* **2022**, *10*, 1055) reported a PP_i sensor that uses lanthanide nanoparticles (LnNP) with a $\text{NaGdF}_4:\text{Nd}$ core exhibiting a single emission band around 1058 nm and showing a high selectivity towards the detection of PP_i with a limit of detection of 3.36 μM . The SWNT sensor presented here exhibits a similar or slightly higher limit of

detection yet lower selectivity towards PP_i . Nevertheless, it represents the first NIR-II sensor for PP_i with an internal calibration for signal readout.

6. Photophysical mechanistic studies on signal transduction are interesting. The authors should comment on how the wavelength shifts of E11 and E11* PL were observed upon [Cu] adsorption.

RESPONSE:

We have addressed this issue in detail in our response to Reviewer 1, Comment 3 (see above) and **added a revised discussion on this topic in the revised manuscript on page 12/13.**

7. The authors should note the difference in the spectral shape between Figs 5e,f and others. Using a different spectrometer won't impact the spectral shapes if both are correctly calibrated. The concentration may not significantly impact the peak shapes as well if the PL spectra were measured in the dilute regime (Powell, J Phys Chem Lett 2018)

RESPONSE:

We thank the reviewer for pointing this out and agree that the difference does not originate from the use of a different spectrometer or SWNT concentration. The note regarding the different spectrometer and lower SWNT concentration was intended to help the reader understand (1) why a different x-axis was chosen (due to the use of a different diffraction grating) and (2) why a stronger response in PL ratio was observed at a concentration of 188 μM , which is not apparent when comparing it to the values presented in Fig. 5b. Peak broadening is sometimes observed at high diazonium salt concentrations and associated with the increased introduction of a second further red-shifted defect emission band (see *Chem. Mat.* **2019**, *31*, 6950-6961). **We clarified our statement in the figure caption of Fig. 5.**

Reviewer #3 (Remarks to the Author):

In this manuscript, the authors describe a biosensor for pyrophosphate (PP) based on sp³-functionalized single-walled carbon nanotubes (SWNTs). The SWCNTs were engineered to reversibly complex copper ions, resulting in a quenching of fluorescence that would be recovered in the presence of PP. The authors further explore the underlying mechanism and application of these sensors to cell lysate and PCR.

There are only a few examples that use sp³-defect fluorescence for optical SWNT sensing. These sensors hold great promise for improving SWNT sensor response and designing next-generation optical sensors more rationally. The manuscript is very well written, with all important controls considered. It lacks only a few details on the description and analysis. I therefore recommend publishing with minor revisions.

Request for additional information:

1. In the introduction, the authors focus on the advantage of better PLQY for sp³-defected SWNTs. It would further motivate this study to mention potential applications for multimodal sensing. The sp³ defect not only brightens SWNTs but also forms new peaks at different wavelengths that could be useful for multi-analyte detection.

RESPONSE:

We thank the reviewer for pointing this out and **made this motivation clearer on page 3.**

2. The authors should specify the physiologically relevant concentration of PP in biological systems.

RESPONSE:

We have addressed this issue in detail in our response to Reviewer 1, Comment 1 (see above) and **added discussion on this topic in the revised manuscript on page 19.**

3. In the experiment of copper-induced fluorescent quenching, both E₁₁ and E₁₁* peaks show a decrease in PL. The authors should compare the extent of E₁₁ quenching to that of unmodified (6,5) SWNTs. This comparison could be useful in understanding the underlying mechanism as to whether the quenching of the E₁₁ peak is entirely due to Cu or if it is affected by the E₁₁* peak.

RESPONSE:

The current understanding of luminescent *sp*³-defects implies that E₁₁* emission occurs upon trapping of E₁₁ excitons at the corresponding defect site without any additional effects. While thermally induced detrapping of E₁₁* excitons and regeneration of E₁₁ excitons is widely assumed to occur at ambient temperature, its impact on overall E₁₁ emission is low (*ACS Nano* 2018, 12, 6, 6059–6065). For example, Berger *et al.* showed that when E₁₁* excitons are quenched, the E₁₁ emission remains stable (*ACS Nano* 2021, 15, 5147–5157). Reduction of E₁₁ emission can be primarily attributed to quenching by Cu²⁺ ions. We performed quenching experiments with pristine SDS-dispersed (6,5) SWNTs and compared the obtained results to the quenching behaviour of functionalized (6,5) SWNTs containing a benzene moiety and alkynyl moiety. Quenching still occurs for pristine (6,5) SWNTs to a similar degree as observed for functionalized SWNTs with the benzene moiety (see below, Fig. 10). Thus, we consider quenching of E₁₁ excitons to be independent of E₁₁* excitons. The stronger quenching effect for functionalized SWNTs with alkynyl moiety likely results from additional coordination of [Cu] to the alkynyl group, which facilitates adsorption to the SWNT surface.

We added this information in the revised Supplementary Information as new Fig. 10.

New Supplementary Fig. 10. **a** PL spectrum of pristine (6,5) SWNTs after the addition of various concentrations of [Cu]. **b** Extracted E₁₁ PL intensities normalized to its initial value before addition of [Cu] of pristine, benzene-functionalized and 4-ethynyl functionalized (6,5) SWNTs vs. concentration of [Cu].

4. The authors propose a hypothesis for the quenching, but little discussion on the shifting. It would strengthen the manuscript to propose a hypothesis (or several hypotheses) for the shifting response (for example, possible conformation effects?), even if this effect is more elusive to study.

RESPONSE:

We have addressed this issue in detail in our response to Reviewer 1, Comment 3 (see above) and **added a discussion on this topic in the revised manuscript on page 12/13.**

Recommendations on tables and figures:

5. Fig 4f is a bit confusing. I suggest redrawing this figure with several equations (such as to indicate binding and release) in the discussion to explain this process more clearly.

RESPONSE:

We have redrawn Figure 4f to clarify the different binding processes. The general concept of Cu^{2+} binding and release is already displayed in Fig. 1.

6. SI Fig 1 is quite important, and I would recommend to move it to the main manuscript.

RESPONSE:

We understand the concern of the reviewer; however, we prefer Figure 1 to remain as it is to clearly focus on the PP_i sensing mechanism, which is fundamental to this study.

7. Fig. 2b caption should clarify that the plot is normalized to the E11 peak intensity.

RESPONSE:

We clarified the normalization to the E₁₁ peak intensity in the figure caption of Fig. 2b in the revised manuscript.

8. In Fig 2f, I suggest directly using peak shift rather than peak position.

RESPONSE:

We understand the concern of the reviewer; however, we prefer the use of peak position to highlight its correlation with Fig 2c and to keep it consistent.

9. In Fig 5e and 5f, the wavelength of the axis should be the same as 5a and 5b.

RESPONSE:

We agree that the choice of a different wavelength scales on the x-axis can be confusing. However, as the displayed PL spectra were measured with a different spectrometer and thus diffraction grating, showing empty data space would not be helpful (see Figure below). We clarified the necessity for a different x-axis in the figure caption of Fig. 5 e,f in the revised manuscript.

Figure 5e with adjusted x-axis.

10. Table S1 should list all the structures of the analytes to help identify possible relationships between some functional groups and responses.

RESPONSE:

We added the chemical structures of all analytes in Table S1 in the revised Supplementary Information.

11. Corrections recommended for a few minor typos: “into an aqueous dispersions” (singular vs. plural) and spacing needed for “of>23uM”

RESPONSE:

We corrected these errors in the revised manuscript.

Reviewer #4 (Remarks to the Author):

Reviewer #5 (Remarks to the Author):

Article Review: Ratiometric fluorescent sensing of pyrophosphate with sp^3 -functionalized single-walled carbon nanotubes

Noteworthy Results:

- The immobilization and displacement of Cu^{2+} ions on sp^3 functionalized (6,5) SWNTs serve as effective near-infrared luminescence probes. Fluorescent quenching occurs when Cu^{2+} ions are immobilized near the SWNT and is restored when the ions are displaced, facilitating the detection of Cu^{2+} complexing analytes. This sensor exhibits some preference for pyrophosphate (PPi), a molecule integral to various biological processes including DNA synthesis and cellular metabolism.
- Diazonium salt covalent functionalization introduces a quantum defect, enabling a ratiometric sensor reading. The sensor exhibits spectral modulation (wavelength shift and PL quenching/enhancement) upon exposure to the target analyte, with a more pronounced response in the E11* peak than the E11 peak. The presence of the E11 peak serves as an embedded reference within the sensor.
- The sensor retains its functionality when transitioning the non-covalently bound corona from SDS to PL-PEG5000. The latter's biocompatibility enables the detection of analytes in biological environments. Consequently, this adaptation of the sensor facilitates the monitoring of PPi in cell lysate during a polymerase chain reaction.

Originality: High

- The sensing scheme is novel, integrating both organic color centers (quantum defects) and Cu^{2+} ion immobilization to achieve sensing, a combination not previously reported to my knowledge.

Broad Appeal: Moderate

- This study could motivate further research exploring innovative ligand systems and metal-ion immobilization techniques for diverse applications.

Clarity: Top 10%

- Incorporating an explanation of amplitude average lifetime prior to introducing its results would enhance those results' comprehensiveness.

RESPONSE:

We agree with the reviewer and included a brief explanation of the amplitude-averaged lifetime in the revised manuscript on page 16.

Revisions:

1) Page 8, lines 172-173: The authors attribute the observed drop in photoluminescence, resulting in the final data point's deviation from the trend, to particle aggregation at elevated analyte concentrations. To substantiate this assumption, it would be good to measure the particle size distribution correlating with varying analyte concentrations.

RESPONSE:

The reduction of PL intensity of SDS-dispersed SWNTs at elevated salt concentrations is a well-known effect. Niyogi *et al.* interpreted this effect as aggregation due to van der Waals interactions that can occur at high counter ion concentrations when the SWNT surface is completely neutralized (*J. Am. Chem. Soc.* **2007**, *129*, 1898–1899). Furthermore, above a certain threshold (0.5 M for NaCl), SDS is no longer soluble in water (*J. Phys. Chem.* **1980**, *84*, 744–751). The combination of both effects, neutralization of SWNT surface and lower solubility of SDS at higher salt concentrations can lead to aggregation effects. Similar studies, but not for SDS-dispersed SWNTs, showed that high concentrations (>0.1 mM) of phosphates can lead to similar results (*Langmuir* **2014**, *30*, 10899–10909). Hence, we doubt that particle size measurements would provide significant insights into this effect. However, as part of Comment 9 by Reviewer 1, **we did perform additional reference experiments for 4-ethynylbenzene functionalized (6,5) SWNTs in the presence of various concentrations of PP_i**. The results agree with our initial assumption of SWNT aggregation. This does not affect the use of SWNTs for sensing because the mentioned effects are observed only high concentrations, which are not relevant for application.

2) Page 17, lines 354-358: The authors postulate that the reduction in PL quenching, observed when comparing SDS to PL-PEG5000, could be attributed to variations in SWNT surface coverage. To strengthen this assertion, the authors should either cite literature that supports this claim or employ an experimental technique to actually measure such surface coverage. A technique like Molecular Probe Absorption has been previously used to quantify the surface coverage of the non-covalently absorbed corona on the carbon nanotube surface (<https://doi.org/10.1021/acs.nanolett.9b02647>). The referenced technique has previously been utilized to actually measure the surface coverage of SDS functionalized SWNTs, the results placing it in the middle of surface coverage when compared to the other reported SWNT coronas; as such it is hard to tell without experiments or literature whether PL-PEG5000 will have more SWNT surface coverage.

RESPONSE:

Based on the analysis of solvatochromic shifts for different SWNT/polymer coatings, Bisker *et al.* (*Nat. Commun.* **2016**, 7, 10241) ranked the surface coverage of PL-PEG₅₀₀₀ among the highest of all studied polymers or surfactants. Similar results were obtained by Polo *et al.* (*J. Phys. Chem. C* **2016**, 120, 3061). Both studies estimated the surface coverage for PL-PEG₅₀₀₀ to be higher than the coverage of single-stranded DNA (AT)₁₅, which shows a higher surface coverage than SDS in the cited paper by the reviewer (Park *et al.*; *Nano Lett.* **2019**, 19, 7712-7724). **We cited the corresponding publications on page 18 in the revised manuscript.**

To further corroborate that the surface coverage correlated with the observed quenching behaviour, we performed a reference experiment with deoxycholate (DOC)-coated (6,5) SWNTs (see below). DOC is known to exhibit a much denser packing on SWNTs than SDS (*J. Phys. Chem. B* **2014**, 118, 6288). Upon addition of 15 μM [Cu], we observe almost no Cu²⁺ induced quenching, clearly demonstrating the correlation between Cu²⁺ ion induced quenching and SWNT surface coverage. **We added this information on page 18/19 in the revised manuscript and added a new Supplementary Fig. 17.**

New Supplementary Fig. 17. PL spectra of 4-ethynylbenzene-functionalized (6,5) SWNTs dispersed in DOC before (black) and after (red) addition of 15 μM [Cu].

REVIEWERS' COMMENTS

Reviewer #1 (Remarks to the Author):

The authors have addressed all my comments.
I am happy to recommend publication.

Reviewer #2 (Remarks to the Author):

All the reviewer's concerns and comments are properly addressed. The paper is publishable in Nature Communications without further review.

Reviewer #3 (Remarks to the Author):

The authors addressed all the critical points. I have no further comments and recommend publication.

Reviewer #4 (Remarks to the Author):

Reviewer #5 (Remarks to the Author):

Thank you for the opportunity to review the revised version of your manuscript. I appreciate the efforts you have made to address the comments and suggestions from the initial round of review.

Upon examination of the revisions, I note that the concerns and recommendations I raised have been adequately addressed.